# The evolution of isolated cavities and hydraulic connection at the glacier bed – Part 1: Steady states and friction laws

**Christian Schoof**

Department of Earth, Ocean and Atmospheric Sciences, University of British Columbia, Vancouver, BC, Canada

**Correspondence:** Christian Schoof (cschoof@mail.ubc.ca)

**Abstract.** Models of subglacial drainage and of cavity formation generally assume that the glacier bed is pervasively hydraulically connected. A growing body of field observations indicates that this assumption is frequently violated in practice. In this paper, I use an extension of existing models of steady-state cavitation to study the formation of hydraulically isolated, uncavitated, low-pressure regions of the bed, which would become flooded if they had access to the subglacial drainage system. I also study their natural counterpart, hydraulically isolated cavities that would drain if they had access to the subglacial drainage system. I show that connections to the drainage system are made at two different sets of critical effective pressure, a lower one at which uncavitated low-pressure regions connect to the drainage system and a higher one at which isolated cavities do the same. I also show that the extent of cavitation, determined by the history of connections made at the bed, has a dominant effect on basal drag while remaining outside the realm of previously employed basal friction laws: changes in basal effective pressure alone may have a minor effect on basal drag until a connection between a cavity and an uncavitated low-pressure region of the bed is made, at which point a drastic and irreversible drop in drag occurs. These results point to the need to expand basal friction and drainage models to include a description of basal connectivity.

## 1  Introduction

Subglacial drainage is often assumed to occur in part through a "distributed" drainage system: connected conduits that are not arborescent in their geometry (Fountain and Walder, 1998) and therefore do not localize drainage into a few large channels (Hewitt, 2010; Schoof et al., 2012; Hewitt, 2013; Werder et al., 2013; Rada and Schoof, 2018; Flowers, 2015). A frequently used paradigm for a distributed drainage model is that of linked cavities (Lliboutry, 1968; Kamb, 1987; Fowler, 1987): localized areas of ice–bed separation in the lee of bed bumps.

Large-scale models for subglacial drainage systems typically assume that the bed as a whole always remains hydraulically connected. Existing process-scale models for the evolution of subglacial cavities generally make the same assumption. In large-scale drainage models, cavities are represented by a water sheet thickness: a cavity depth averaged over a representative small area of the bed (that is, an area of the bed that is much larger than an individual cavity but much smaller than the glacier as a whole). The assumption of a connected bed here simply means that water can flow as soon as the sheet thickness exceeds zero (e.g. Werder et al., 2013; Sommers et al., 2018). As a result, local variations in water pressure at the scale of individual cavities are small, since they would otherwise lead to excessive water fluxes, and water pressure is a well-defined, smoothly varying variable in the large-scale model.

In process-scale models, hydraulic connectedness typically occurs through the bed itself: the bed is highly permeable. Water sourced from an ambient drainage system at some given water pressure can force its way between the ice and bed as soon as compressive normal stress at the base of the ice drops to the water pressure in the ambient drainage system, causing a cavity to form (Schoof, 2005; Gagliardini et al., 2007; Helanow et al., 2020, 2021; Stubblefield et al., 2021; de Diego et al., 2022, 2023).

These assumptions are at odds with a growing set of observations (Hodge, 1979; Murray and Clarke, 1995; Andrews

et al., 2014; Lefeuvre et al., 2015; Rada and Schoof, 2018) indicating that hydraulic connections at the glacier bed are often patchy and evolve in time: while the bed itself may be somewhat permeable, that permeability is too low to allow significant water transport on the timescales over which the drainage system evolves. On these timescales, water must then flow predominantly along the ice–bed interface, and the topology of the conduit network present there (consisting of subglacial cavities and other forms of void space like R-channels) may not provide a connection to all parts of the bed.

Recent work in large-scale drainage modelling has attempted to address this issue (Hoffman et al., 2016; Rada and Schoof, 2018), albeit in fairly crude form: for instance, one possibility is to assume that water can only flow when sheet thickness exceeds a critical value. The aim of the present work is to study the evolution of cavities in more detail for an effectively impermeable bed at the process scale to better understand how an ambient active drainage system can access other parts of the bed through the evolution of basal cavities. By contrast with most studies of subglacial cavity formation, my focus is mostly on the evolution of subglacial connectivity rather than on the computation of a sliding law. As a by-product, I also show that connectivity plays a major role in controlling friction at the glacier bed.

If only part of the glacier bed has access to the ambient drainage system, then isolated, uncavitated, low-pressure regions can form elsewhere, at normal stresses that would lead to ice–bed separation if water from the ambient drainage system had access. Conversely, these distant parts of the glacier bed can become flooded with water when connected cavities grow at low effective pressure. If the effective pressure in the drainage system increases again after that flooding, the intervening connections can become closed, leaving isolated cavities of fixed volume. These isolated cavities will generally be at different effective pressures than the connected drainage system.

In the present work, I have used a modified mathematical model for cavity formation to explore the physics involved. The basic physics of ice flow over an undulating bed, allowing for the possibility of ice–bed separation as water forces its way between the two, are the same as in existing models for subglacial cavity formation. However, only a pre-defined, highly permeable part of the bed, denoted by $P$, is assumed to be directly connected to the ambient drainage system: as in the existing models of de Diego et al. (2022, 2023), Gagliardini et al. (2007), Helanow et al. (2020, 2021), Schoof (2005), and Stubblefield et al. (2021), water is assumed to force its way between the ice and bed if compressive normal stress on $P$ drops to the value of the water pressure in the ambient drainage system. The remainder of the bed is assumed to be completely impermeable. Water can access these other parts of the bed interface (outside of $P$) only if there is a hydraulic connection to $P$ along the ice–bed interface. Moreover, if water has previously accessed some impermeable part of the bed and the hydraulic connection has subsequently been closed, then an isolated cavity is formed. The water pressure in that isolated cavity can differ from the water pressure in the ambient drainage system, but the volume of the cavity will remain fixed.

The model comes in two flavours: first, a two-dimensional, purely viscous flow model for the ice assumes that the cavity roof is in steady state and that water pressure in each separate cavity is spatially uniform. Where a cavity is in contact with the permeable part $P$ of the bed, water pressure equals that in the ambient drainage system, while water pressure in isolated cavities is dictated by their volume. Second, a more general dynamic model assumes viscoelastic ice flow and explicitly considers how water is redistributed within the cavities by water pressure gradients, in a manner analogous to hydrofracture models for pre-existing cracks. The hydraulic conductivity that controls water flow is large within cavities (ensuring rapid equilibration) but vanishes when the ice–bed gap is zero, thereby allowing the model to capture the formation of isolated cavities and of isolated but uncavitated low-pressure regions in a dynamic framework.

The two versions of the model are susceptible to solution by different methods, making the simpler, purely viscous, steady-state version a useful test case for the more complicated dynamic version. To make the presentation more manageable, I have split these two model versions across two separate papers, focusing here on the purely viscous steady-state model. The dynamic model is presented in a companion paper (Schoof, submitted), which I will refer to below as Part 2. The present paper is structured as follows: first, I describe the mathematical model formulation in Sect. 2, with various technical aspects of the solution relegated to the appendices. In Sect. 3.1, I investigate how cavity extent depends on the effective pressure in the ambient drainage system, on the location of the permeable part $P$ of the bed directly connected to the ambient drainage system, and on the past history of cavity formation across the bed. Subsequently, I use these solutions for cavity geometry in Sect. 3.2 to compute friction laws: that is, the corresponding amount of basal drag as a function of sliding velocity and effective pressure. I then investigate in Sect. 3.4 whether changes in bed geometry qualitatively affect the results. Implications for large-scale models of subglacial hydrology and glacier dynamics are discussed in Sect. 4.

## 2 A two-dimensional viscous steady-state model

Consider the possibility of isolated cavities in the two-dimensional, purely viscous, steady-state model of subglacial cavitation in Fowler (1986) and Schoof (2005). Based on the approximation of small bed slopes pioneered in Nye (1969) and Kamb (1970), the model can be written as follows: ice occupies the half-space $y > 0$ in the Cartesian $(x, y)$ plane. In that domain, ice flow satisfies Stokes' equa-

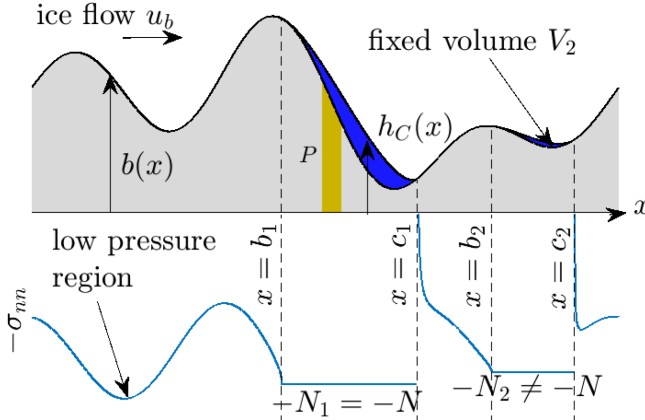

**Figure 1.** Definitions used in the model. The upstream and downstream cavity end points of the $j$th cavity are denoted by $b_j$ and $c_j$, respectively. $a$ is the width of the periodic domain. $h(x)$ is the cavity roof height and $b(x)$ the local bed elevation. I use beige colouring throughout the paper to indicate the permeable part $P$ of the bed and grey for the impermeable part. The blue curve $-\sigma_{nn} = p - 2\eta \partial v/\partial y$ shows that compressive normal stress against $x$. $-\sigma_{nn}$ must exceed the negative effective pressure $-N_j$ locally around any given cavity $j$ as shown, but not globally, allowing low-pressure contact areas (with normal stress below $-N$) to exist as shown. Cavity $j = 2$ here is an isolated cavity, with fixed volume $V_2$, while cavity $j = 1$ is connected as it overlaps with $P$.

tions,

$$\eta \nabla^2 \boldsymbol{u} - \nabla p = \boldsymbol{0}, \quad \nabla \cdot \boldsymbol{u} = 0. \tag{1}$$

Here, $\boldsymbol{u} = (u, v)$ is the perturbation in ice velocity around a mean $(u_b, 0)$ introduced by flow over bed topography, while p is the reduced pressure (that is, the actual pressure minus the cryostatic overburden), $\nabla$ is the usual two-dimensional gradient operator, and $\eta$ is the viscosity of ice, assumed to be constant here, and I assume that $u_b > 0$ so mean flow is to the right in Fig. 1.

To be definite, I also assume the domain to be periodic in $x$ with period $a$ (Fig. 1). At the base of the ice $y = 0$, let the set of points at which there is contact between ice and bed be denoted by $C'$, and let the complement $C$ denote cavities, or regions of ice–bed separation. **TS1** For $x \in C'$, the normal component of velocity vanishes, leading to the boundary condition

$$v = u_b \frac{\partial b}{\partial x}, \tag{2}$$

where $b(x)$ is the elevation of the bed about a mean. Conversely, let $C$ be composed of a set of disjoint intervals $C_j = (b_j, c_j)$, each representing a separate cavity. On each $C_j$, normal stress is prescribed in the form

$$p - 2\eta \frac{\partial v}{\partial y} = -N_j, \tag{3}$$

where $N_j$ is the effective pressure in the $j$th cavity, defined as the difference between overburden and water pressure in the cavity. The cavity roof elevation $h_C$ satisfies the steady-state kinematic boundary condition

$$v = u_b \frac{\partial h_C}{\partial x} \tag{4}$$

on $C$ with $h_C = b$ at cavity end points, so the lower boundary of the ice is continuous. These boundary conditions are combined with far-field conditions:

$$p, u \to 0 \quad \text{as } y \to \infty. \tag{5}$$

The previous work in Schoof (2005) assumed that the water pressure in each cavity is the same, implicitly requiring a highly permeable bed and allowing a universal effective pressure to be defined as $N = N_j$ for all $j$. Taking the implied permeability of the bed further, Schoof (2005) added the inequality constraints

$$p - 2\eta \frac{\partial v}{\partial y} \geq -N \quad \text{for } x \in C' \tag{6}$$

$$h_C > b \quad \text{for } x \in C \tag{7}$$

in order to determine the extent of cavities. Physically, these inequalities represent the idea that normal stress cannot be less than the (assumed uniform) water pressure anywhere at the bed, since water will force its way between the ice and bed in that case, forming a new cavity, and that a cavity only exists if the cavity roof is indeed above the bed.

Here I abandon the assumption of a fully permeable bed. If parts of the bed are instead impermeable, there is no universally defined water pressure, and water will not force its way between the ice and bed simply because the normal stress drops locally to the water pressure in a distant drainage system. Water pressure is still assumed to be constant in each cavity while potentially differing between cavities, so the $N_j$ values are constants but need not be equal to one another. As a result, the constraint (6) also need no longer hold across the bed.

To be more specific, I assume that only a part $P$ of the bed is permeable and connected to a drainage system at prescribed effective pressure $N$. Hence the condition (Eq. 6) holds for $x \in P$, and any cavity straddling a part of $P$ will be "connected" at the drainage pressure $N$. ($P$ is a part of the bed, but specified here only in terms of the horizontal coordinates of points in $P$ at the ice–bed interface, since no depth-dependent physics in the bed are resolved by the model.) Any cavity not straddling $P$ will be "isolated" and required to hold a prescribed volume of water,

$$V_j = \int_{b_j}^{c_j} (h_C - b)\,\mathrm{d}x, \tag{8}$$

which, if a solution exists, determines the effective pressure $N_j$. The constraint (7) still holds, but the inequality (6) is instead replaced by the weaker requirement that

$$p - 2\eta\frac{\partial v}{\partial y} > -N_j \quad (9)$$

in some finite intervals $(b_j - \delta, b_j)$ and $(c_j, c_j + \delta)$ (that is, there is some $\delta > 0$ such that the constraint 9 holds), ensuring that the cavity remains sealed.

Outside of the intervals $(b_j - \delta, b_j)$ and $(c_j, c_j + \delta)$, the inequality (9) can, and in general will, be violated somewhere as indicated in Fig. 1. The possibility of such underpressurized regions is the primary difference between the permeable and impermeable bed models. By not bounding compressive normal stress everywhere at the bed, however, the model does not allow a vapour-filled cavity to form if the normal stress drops to the triple-point pressure of water. In order to incorporate the latter effect, I would need to add the constraint that $p - 2\eta\partial v/\partial y > -p_i$ in $C'$, where $p_i$ is ice overburden, and set $p - 2\eta\partial v/\partial y = -p_i$ in any cavity that does not straddle $P$ and in which the prescribed water volume $V_j$ (potentially equal to zero) would lead to an effective pressure $N_j > p_i$ if the volume constraint (8) were imposed. I omit this complication here on the basis that I expect overburden $p_i$ to be large compared with the typical normal stress variations caused by ice flow over bed undulations; suffice it to point out that the model described in Part 2 can in principle describe vapour-filled cavities.

Also note that Stubblefield et al. (2021) employ a similar but ultimately distinct volume constraint to Eq. (8): theirs is a global constraint, in which the bed is fully permeable (equivalent to $P = (0, a)$ here) and all cavities are at the same effective pressure, but the latter is not prescribed. Instead, the total cavity volume is prescribed through initial conditions, the constraint itself being imposed on normal velocity so as to conserve that initial volume. Equation (8) here is a local constraint instead, prescribing the volume of an individual cavity.

The specification of a permeable bed portion $P$ may be awkward but is realistically the only way to model partial access of the drainage system to the bed in two dimensions. Strictly speaking, water here is assumed to flow *through* the permeable bed in $P$ in order to access connected cavities, but $P$ can also be thought of as locations where an ambient drainage system is able to access the flowline being modelled laterally along the ice–bed interface, with the lateral dimension remaining unresolved. Below, I will typically consider either the entire bed permeable with $P = (0, a)$, or I will consider a small permeable patch around a single location, which I will denote by $x_P$. I will typically choose $x_P$ to be the location of a local minimum of compressive normal stress for an uncavitated bed, since that is where cavities first form for a permeable bed. In addition, in Sect. 3.3 I consider larger permeable bed portions $P$ that do not align with these normal stress minima.

In any case, the modified steady cavity problem can be solved by a slight modification to the complex variable method in Schoof (2005), whose numerical method I also adapt. The technical detail is relegated to the Appendix. A steady-state solution to the model is likely to be highly non-unique, since the placement of prescribed water volumes $V_j$ in isolated cavities is history-dependent and quite arbitrary in a steady-state model (by contrast, the dynamic model described in Part 2 self-consistently determines the volume of isolated cavities, precisely because it tracks the evolution in time of cavities).

In the next subsection, I consider a system of cavities that is in quasi-equilibrium, forced by a very slowly changing effective pressure $N$ in the ambient drainage system. I also assume that the bed starts with no cavities. The latter initially form around the permeable parts $P$ of the bed when $N$ is made sufficiently small. The cavities at first remain trapped between prominent protrusions but can drown these bed protrusions abruptly when $N$ is decreased to some critical values; I describe the method by which I compute the enlarged cavity in detail in Appendix A4. If $N$ is increased again, the extended cavity roof can then make contact again with the drowned bed protrusion, thereby (in two dimensions) sealing the lee side of that protrusion and forming an isolated cavity. The volume of that isolated cavity is dictated by cavity size at the point where the cavity roof re-contacts, making the solution unique for a sequence of slow changes in $N$. Again, Appendix A4 provides further detail.

## 3 Results

### 3.1 Cavity geometry

Figure 2 shows the evolution of cavity geometry for the double-bumped periodic geometry,

$$b(x) = h_0\left[\sin\left(\frac{2\pi x}{a}\right) + \sin\left(\frac{4\pi x}{a}\right)\right], \quad (10)$$

with $h_0$ and $a$ constant. I focus first on the reference case of a fully permeable bed, as previously considered in Fowler (1986), Schoof (2005), Gagliardini et al. (2007), Helanow et al. (2020, 2021), Stubblefield et al. (2021), and de Diego et al. (2022, 2023).

Note that, when expressed as functions of a scaled position $x^* = 2\pi x/a$ along the bed, cavity size and shape depend only on the following dimensionless effective pressure (Fowler, 1981, 1986):

$$N^* = \frac{Na^2}{4\pi^2 h_0\eta u_b}, \quad (11)$$

and I adopt this here to reduce the parameter space to be explored. Similarly, a dimensionless compressive normal stress

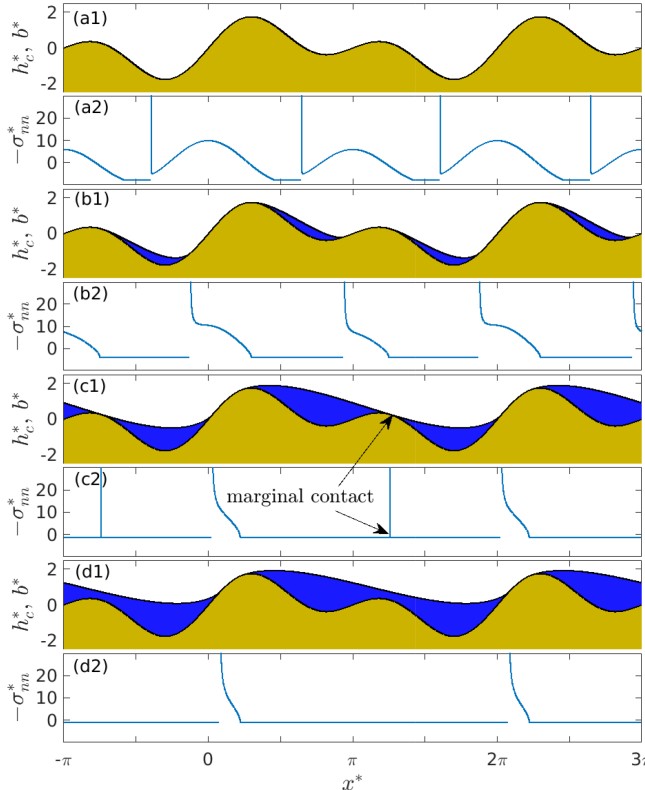

**Figure 2.** Cavity roof shape $h_C^*(x^*)$ and bed elevation $b^*(x^*)$ for the bed given by Eq. (10) with $P^* = (0, 2\pi)$ and **(a1)** $N^* = 7.6$, **(b1)** $N^* = 4.02$, **(c1)** $N^* = 1.19$, and **(d1)** $N^* = 0.91$. The corresponding compressive normal stresses $-\sigma_{nn}^*$ are plotted in panels **(a2)–(d2)**.

defined by

$$-\sigma_{nn}^* = \frac{a^2}{4\pi^2 h_0 \eta u_b} \left( p - 2\eta \frac{\partial v}{\partial x} \right) \bigg|_{y=0} \quad (12)$$

also depends only on $N^*$ when expressed in terms of the scaled position $x^*$. I use $\sigma_{nn}^*$ to visualize normal stresses at the bed. In the same vein, I use $b^* = b/h_0$ and $h_C^* = h_C/h_0$ as scaled bed and cavity roof elevations, and I use $P^* = \{x^* : x^* a/(2\pi) \in P\}$ as the scaled version of the permeable bed.

With the bed geometry given by Eq. (10), the basal compressive normal stress $-\sigma_{nn}^*$ has two equally deep minima around $x^* = 1.64$ and $x^* = 4.65$ prior to cavity formation. Two cavities per bed period form simultaneously around these locations in the lee of the two bed protrusions when effective pressure $N^*$ drops below a critical value $N_{\mathrm{init}}^* = 8.06$ (panel a1 and a2 of Fig. 2). The cavity roof $h_C^*$ remains very close to the bed $b^*$ in the cavities initially, which are therefore easier to discern in the normal stress distribution $-\sigma_{nn}^*$ (panel a2). The pattern of normal stress shown here is common to the steady-state cavity solutions computed elsewhere (Fowler, 1986; Schoof, 2005; Gagliardini et al., 2007; Stubblefield et al., 2021; de Diego et al., 2022, 2023): compres-

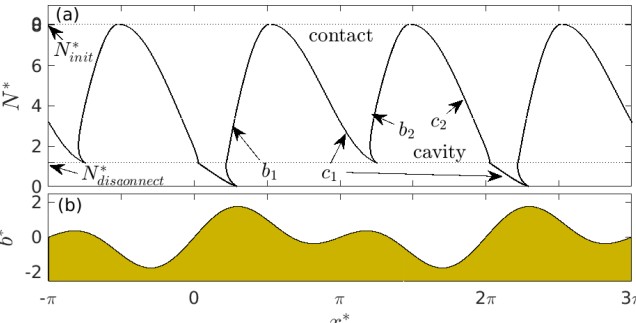

**Figure 3.** Panel **(a)** shows the effective pressure $N^*$ against cavity end-point positions $b_j^*$ and $c_j^*$ for a fully permeable bed of the form in Eq. (10). $N_{\mathrm{init}}^*$ and $N_{\mathrm{disconnect}}^*$ are defined in the main text, and "contact" and "cavity" mark the sides of the black curve occupied by contact areas and cavities. Panel **(b)** shows the corresponding upper surface elevation $b^*(x^*)$ of the bed against $x^*$.

sive normal stress is continuous at the upstream end of the cavity, with larger values immediately outside the cavity than inside acting to contain the water in the cavity, and normal stress has a positive singularity at the downstream cavity end. I show in Appendix A5 that this stress pattern necessarily follows from the inequalities (7) and (9).

The cavities expand continuously as $N^*$ is lowered further until they merge at a second critical value $N_{\mathrm{disconnect}}^* = 1.19$ (panels c1 and c2), and the merged cavity then continues to expand further. If $N^*$ is raised again, cavity evolution is completely reversible: for instance, the merged cavity once more separates in two at $N^* = N_{\mathrm{disconnect}}^*$.

The dependence of cavity size on $N^*$ can be visualized by plotting cavity end-point positions $b_j^* = 2\pi b_j/a$ and $c_j^* = 2\pi c_j/a$ against $N^*$ (Fig. 3, where the two critical values are marked with broken horizontal lines). Note that there is a unique solution for every $N^*$ here, corresponding to either a single merged cavity or two separate cavities. The labels "contact" and "cavity" indicate which side of the black curves corresponds to a contact area and a cavity, respectively. A second key feature of Fig. 3 is that the contact areas disappear at $N^* = 0$ and no solution exists for negative effective pressures $N^* < 0$: naturally, when water pressure is above overburden, force balance can no longer be maintained.

The behaviour is somewhat different if I restrict the permeable portion $P^*$ of the bed to a small region around the upstream stress minimum at $x^* = x_P^* := 1.64$ (Fig. 4). With only this small portion of the bed being permeable, a cavity starts to form at $x_P^*$ when $N^* = N_{\mathrm{init}}^* = 8.04$. As effective pressure in the drainage system is lowered, the cavity grows first on the lee side of the large bed protrusion to the left (to which the cavity is "attached"), while the lee of the smaller protrusion to the right remains uncavitated as shown in Fig. 4a1. (Note that I will use "large" or "prominent" protrusion to describe the protrusion that has the largest differ-

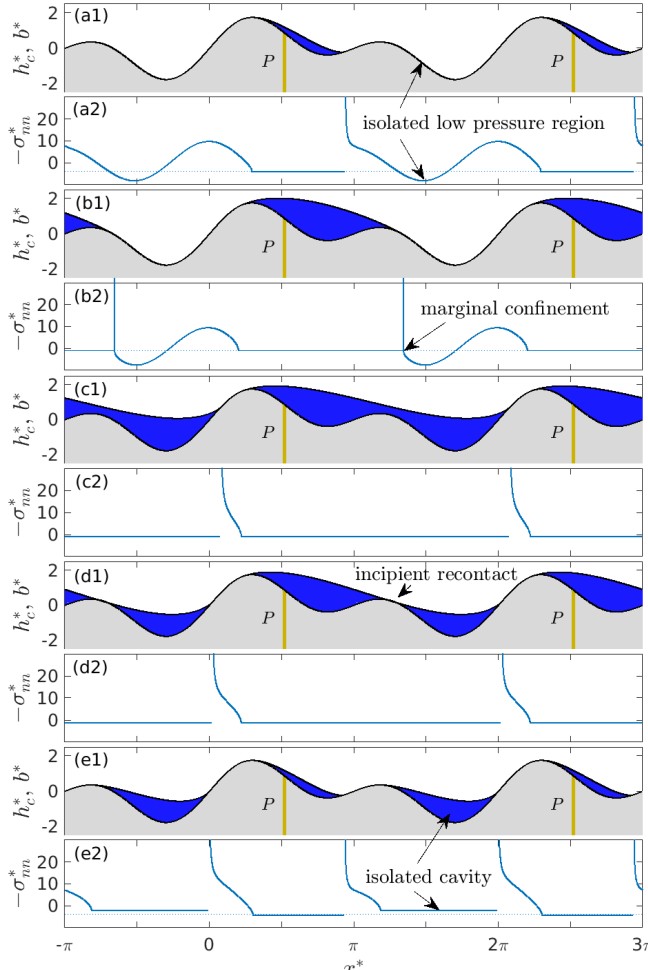

**Figure 4.** Cavity roof shape $h_C^*(x^*)$ and bed elevation $b^*(x^*)$ for the bed given by Eq. (10) with $P^* = \{1.64\}$ and **(a1)** $N^* = 4.01$, **(b1)** $N^* = 0.92$, **(c1)** $N^* = 0.91$, **(d1)** $N^* = 1.19$, and **(e1)** $N^* = 4.02$. The permeable and impermeable portions of the bed are rendered in beige and grey, respectively. The corresponding normal stresses $-\sigma_{nn}^*$ are plotted in panels **(a2)**–**(e2)**; note that the isolated cavity in **(e2)** is at a different constant pressure than the connected cavity around the permeable bed portion $P^*$.

ence in height between the local maximum at its top and the local minimum on its upstream side.) This contrasts with the fully permeable bed case, where the lee sides of both bed protrusions become cavitated at the same $N^*$ (see also Fig. 3).

5 As before, the normal stress around the cavity is continuous at the detachment point at the upstream end of the cavity and singular at the reattachment point at the downstream end (Fig. 4a2). Normal stress exceeds $-N^*$ at both ends as required by the constraint (9). Note, however, that normal 10 stresses on the lee side of the smaller protrusion are lower than $-N^*$, and inequality (6) is violated there, away from the permeable bed portion $P$: an isolated underpressurized region forms here, separated from the cavity by the high-normal-stress region in the lee of the cavity.

As $N^*$ is decreased, the cavity expands, while the size of 15 the high-stress region isolating the lee of the smaller bed protrusion shrinks. Eventually, the confinement of the cavity at its downstream end becomes marginal (Fig. 4b2) at $N^* = N_{\text{connect}}^* = 0.92$. A further reduction in $N^*$ causes the cavity to expand abruptly across the top of the smaller bed 20 protrusion (Fig. 4c).

The newly expanded cavity roof now has a finite size gap above the smaller bed protrusion. It expands further, but now continuously, if effective pressure is lowered again. The expanded cavity is in fact identical in shape to the single merged cavity that forms for a fully permeable bed at 25 the same effective pressure. More significantly, if $N^*$ is increased again from the critical value of $N_{\text{connect}}^*$, the cavity roof does not immediately re-contact the bed again. In order for the enlarged cavity roof to re-contact the smaller bed 30 protrusion, $N^*$ has to increase by a finite amount to $N^* = N_{\text{disconnect}}^* = 1.19 > N_{\text{connect}}^*$ (Fig. 4d). That higher critical value is equal to the effective pressure at the merger of the two cavities that form independently in the lee of both bed protrusions when the entire bed is permeable, $P^* = (0, 2\pi)$ 35 (Fig. 3), and I use the same symbol $N_{\text{disconnect}}^*$ deliberately.

In the present two-dimensional model, re-contact with a limited permeable bed portion immediately leads to the formation of a second isolated cavity downstream of the right-hand bed protrusion, which I treat as retaining a constant 40 volume $V_2 = 1.062 a h_0 / (2\pi)$ after reattachment (this being the volume at reattachment). A further increase in effective pressure $N^*$ in the permeable portion $P^*$ of the bed leads to the original cavity in the lee of the left-hand bed protrusion shrinking again, eventually disappearing at a critical value 45 of $N_{\text{shrink}}^* = 7.99$, slightly less than the value of $N_{\text{init}}^* = 8.06$ at which the cavity was originally formed. Meanwhile, the effective pressure $N_2^*$ in the isolated cavity typically differs from $N_1^* = N^*$ in the connected cavity (Fig. 4e). Note that the solution is non-unique here: panels (a) and (e) of Fig. 4 50 correspond to (nearly) the same effective pressure $N^*$.

Conversely, if $N^*$ is lowered again, the cavity that is attached to the larger bed protrusion on the left will reconnect to the isolated cavity that is attached to the smaller bed protrusion on the right at the same critical value $N_{\text{disconnect}}$ at 55 which the isolated cavity originally formed: changes in cavity geometry become reversible once the lee sides of both bed protrusions have become cavitated.

The dependence of cavity end points on $N^*$ is again plotted systematically in Fig. 5a, which is analogous to Fig. 3. 60 The black curves show cavity end-point positions that result if I start with an uncavitated bed and only lower $N^*$, with the abrupt cavity enlargement at $N^* = N_{\text{connect}}^*$ clearly visible as a discontinuity in the downstream cavity end-point position $c_1$. The upstream cavity end point in fact shifts dis- 65 continuously too, but by an amount that may be too small to discern. As in Fig. 3, the contact area again vanishes at $N^* = 0$ and no solution exists for negative $N^*$: in fact, the solutions in Figs. 3 and 5a are identical for $N^* < N_{\text{connect}}^*$. If,

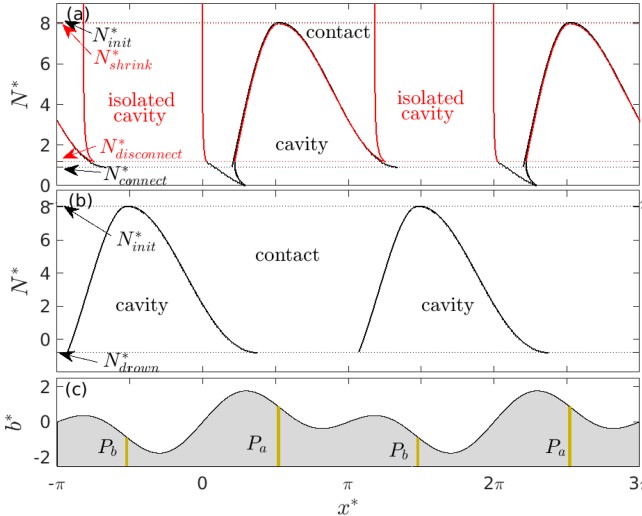

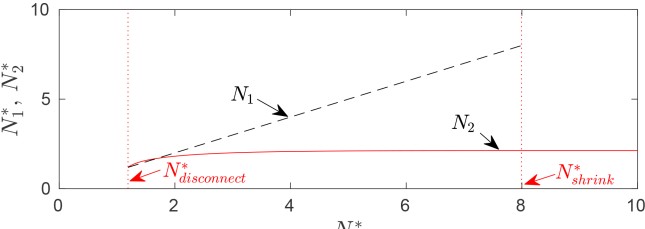

**Figure 6.** In red is the effective pressure $N_2^* = -\sigma_{nn}^*$ in the isolated cavity formed as in Fig. 4d against effective pressure $N^*$ in the connected cavity around the permeable bed portion $P$. The effective pressure $N^* = N_1^*$ in the connected cavity is plotted as a black dashed line, terminated at $N^* = N_{\text{shrink}}^*$, where the connected cavity disappears. The isolated cavity exists past $N_{\text{shrink}}^*$, but not for $N^* < N_{\text{disconnect}}^*$.

**Figure 5.** Panel **(a)** shows effective pressure $N^*$ against cavity endpoint positions $b_j^*$ and $c_j^*$ for a bed of the form in Eq. (10) with $P^*$ concentrated around 1.64. $N_{\text{init}}^*$, $N_{\text{shrink}}^*$, $N_{\text{disconnect}}^*$, and $N_{\text{connect}}^*$ are defined in the main text, and "contact" and "cavity" mark contact areas and cavities on either side of the black curves (solutions obtained by starting with an uncavitated bed at $N^* = N_{\text{init}}^*$ and lowering $N^*$). The red curves show solutions obtained when $N^*$ is lowered below $N_{\text{connect}}^*$ and raised above $N_{\text{disconnect}}^*$ subsequently. The newly formed isolated cavity is marked in red. Panel **(b)** shows effective pressure $N^*$ against cavity end-point positions $b_j^*$ and $c_j^*$ for a bed of the form in Eq. (10) with $P^*$ concentrated around $x_P^* = 4.65$. $N_{\text{drown}}^*$ is defined in the main text. Panel **(c)** shows the corresponding elevation $b^*(x^*)$ of the upper surface of the bed against $x^*$. The beige strips labelled $P_a$ and $P_b$ indicate the permeable bed portions used in panels **(a)** and **(b)**, respectively.

on the other hand, $N^*$ is first lowered below $N_{\text{connect}}^*$ and raised again, the cavity end-point solution follows the red curve above the disconnection value $N_{\text{disconnect}}^*$. Note that the isolated cavity that forms (indicated by red lettering) initially
shifts slightly upstream as $N^*$ is increased above $N_{\text{disconnect}}^*$, but then remains relatively unaltered as the connected cavity shrinks and disappears.

In addition, I have plotted the effective pressure $N_2^*$ in the isolated cavity against the forcing effective pressure $N^*$ in
Fig. 6. The effective pressure $N_2^*$ mostly increases as $N^*$ does, implying a drop in water pressure in the isolated cavity as water pressure in the connected drainage system drops, albeit at a slower rate. This may be surprising given observations of anticorrelated water pressures between connected
and unconnected parts of the bed (Murray and Clarke, 1995; Lefeuvre et al., 2015; Rada and Schoof, 2018). There are two important differences here: first, the water pressure variations being considered are not transient, and consequently the size of both cavities has fully adjusted to steady-state conditions
after a change in effective pressure $N^*$. Second, in a flowline model, the redistribution of normal stress considered by Murray and Clarke (1995) and Lefeuvre et al. (2018) is mod-

ulated by flow over bed topography and by changes in the extent to which bed topography is drowned by cavities. For the bed geometry in Eq. (10) under consideration, an increase
in $N^*$ in the connected cavity leads to more of the upstream face of the right-hand protrusion being covered by ice. The need to flow up and over that protrusion leads to a reduction in normal stress in its lee and hence to a drop in the water pressure required to maintain a cavity of fixed volume in its
lee. This explains the increase in $N_2^*$ with increases in $N^*$ here.

The ability of a cavity to expand across bed protrusions and subsequently create isolated cavities as described above depends on the position of the permeable portion of bed rel-
35 ative to prominent bed protrusions. Consider the same bed given by Eq. (10), but move the permeable portion of the bed to $x^* = 4.65$ [TS2]. In that case, a cavity initiates here at the same initial value $N_{\text{init}}^* = 8.06$. Now, however, the cavity is attached to a smaller bed protrusion and remains confined in
its lee for all positive values of $N$, separated from the low-pressure region downstream of the more prominent protrusion by high normal stresses on either side of the cavity. This confinement in fact persists all the way to a *negative* effective pressure $N_{\text{drown}}^* = -0.79$ (Fig. 7). Beyond this critical effec-
tive pressure, the ice fully detaches from the bed, and vertical force balance is once more violated.

Note that the cavity is not able to expand upstream to the lee side of the bigger bed protrusion and only expands downstream past that bigger protrusion at the negative effective
pressure $N_{\text{drown}}^*$ when the confinement at the downstream end disappears: the normal stress upstream of the cavity remains in excess of water pressure even then. As with the previous example, I have plotted the position of cavity end points against $N^*$ in Fig. 5b. As in the fully permeable bed case in
Fig. 3, there is now a unique solution, though it no longer disappears at $N^* = 0$. Note that the ability of cavities to remain contained at negative effective pressure due to uneven stresses induced by ice flow over topography may also be significant observationally, since sustained negative effective

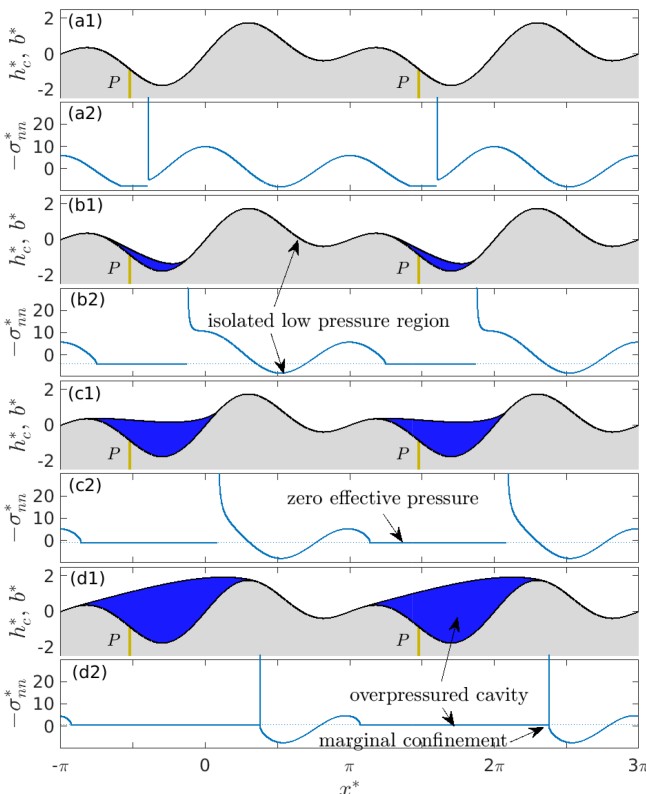

**Figure 7.** Cavity roof shape $h_C^*(x^*)$ and bed elevation $b^*(x^*)$ for the bed given by Eq. (10) with $P^* = \{4.65\}$ and **(a1)** $N^* = 7.60$, **(b1)** $N^* = 4.02$, **(c1)** $N^* = 0$, and **(d1)** $N^* = -0.79$. The permeable and impermeable portions of the bed are rendered in beige and grey, respectively. The corresponding normal stresses $-\sigma_{nn}^*$ are plotted in panels **(a2)**–**(d2)**; note that positive values of $-\sigma_{nn}^*$ as in panel **(d2)** correspond to negative effective pressure.

pressures are a frequent feature of borehole water pressure records (e.g. Rada and Schoof, 2018).

### 3.2 Basal drag

We can also ask how the formation of isolated cavities, and confinement of cavities, affects basal drag defined through (Fowler, 1986; Schoof, 2005)

$$\tau_{\mathrm{b}} = \frac{1}{a} \int_0^a \left( p - 2\eta \frac{\partial v}{\partial x} \right) \bigg|_{y=0} \frac{\partial h_C}{\partial x} \mathrm{d}x, \tag{13}$$

where I treat $h_C = b$ in the contact areas $C'$. As above, this can be cast in dimensionless form, now defining

$$\tau_{\mathrm{b}}^* = \frac{\tau_{\mathrm{b}} a}{2\pi h_0 N}, \tag{14}$$

which is then a function of $N^*$ only (Fowler, 1986). For consistency with Fowler (1986), Schoof (2005), and Gagliardini et al. (2007), I plot $\tau_{\mathrm{b}}^*$ against $1/N^* = 4\pi^2 h_0 \eta u_b/(a^2 N)$ to visualize the resulting friction law, with $1/N^*$ effectively

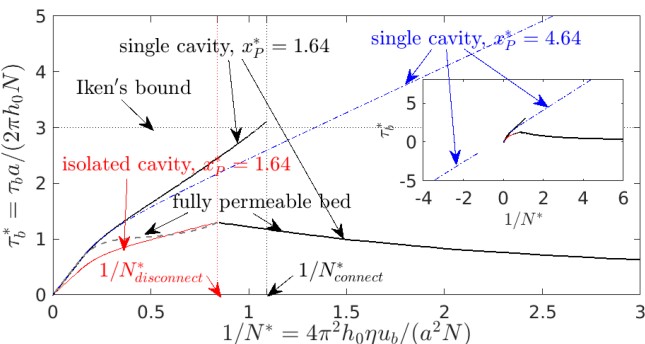

**Figure 8.** Friction law for the bed given by Eq. (10): $\tau_{\mathrm{b}}^* = \tau_{\mathrm{b}} a/(2\pi N)$ against $1/N^* = 4\pi^2 h_0 \eta u_b/(a^N)$ for the solutions shown in Figs. 3 and 5. The solid black curve (consisting of multiple segments) corresponds to the solution shown as a black curve in Fig. 5a (single cavity connected to a permeable bed $P^*$ around $x^* = 1.64$), while the red curve here also corresponds to the red curve in Fig. 5a (connected cavity around $x^* = 1.64$ and an isolated cavity around $x^* = 4.65$). The dashed blue curve corresponds to the solution in Fig. 5b (single connected cavity around $x^* = 4.65$); because the latter solution extends to negative $N^*$, the friction law can likewise be extended to values of $1/N^* < 1/N_{\mathrm{drown}}^* < 0$ as shown in the inset. The continuous dashed black curve (partly obscured by the solid black curve for $1/N^* > 1/N_{\mathrm{connect}}^*$) corresponds to the solution for a fully permeable bed in Fig. 3. The line labelled "Iken's bound" is at $\tau_{\mathrm{b}}^* = \max(\partial b^*/\partial x^*)$.

being a proxy for ice velocity $u_b$. Results for the double-humped bed given by Eq. (10) are shown in Fig. 8.

The standard assumption of a fully permeable bed $P^* = (0, 2\pi)$ gives rise to the single-valued, continuous black dashed curve (partly obscured by the solid black curve as indicated by the arrow marked "fully permeable bed"). It corresponds to relatively small values of $\tau_{\mathrm{b}}^*$ that satisfy Iken's bound $\tau_{\mathrm{b}}^* \le \max(\partial b^*/\partial x^*)$ (Schoof, 2005): the maximum possible basal drag that can be attained is bounded by bed slope, where, with the bed shape given by Eq. (10), $\max(\partial b^*/\partial x^*) = 3$. The shape of the dashed black curve mirrors some of those in Schoof (2005).

With a small $P^*$ centred around $x_P^* = 1.64$, the friction law changes significantly: the relationship of $\tau_{\mathrm{b}}^*$ [TS3] comes in multiple branches, depending on the presence of isolated cavities. When there is only a cavity in the lee of the prominent bed protrusion on the left, basal drag is quite high and can exceed Iken's bound (whose derivation in Schoof, 2005, is based on a permeable bed). Drag $\tau_{\mathrm{b}}^*$ drops abruptly when $N_{\mathrm{connect}}^*$ is reached and the cavity expands to drown out not only the second smaller bed protrusion, but also a significant part of the steeper slope behind it (solid black curve). Once the cavity has expanded and $N^*$ is increased again, an isolated cavity forms, leading to values of $\tau_{\mathrm{b}}^*$ that are generally comparable to those for a fully permeable bed (solid red curve). Below $1/N_{\mathrm{shrink}}^* \approx 0.125$, $\tau_{\mathrm{b}}^*$ then simply becomes linear in $1/N^*$: this implies that $\tau_{\mathrm{b}} \propto u_b$ independent of $N$,

as is familiar from theories of basal sliding in the absence of cavitation (Nye, 1969; Kamb, 1970). In the absence of expanding or shrinking connected cavities, an isolated cavity simply adopts a constant shape and changes its internal water pressure to keep that shape. The effect of such a constant-shape isolated cavity on steady-state basal drag is the same as for a rigid bed, since the shape of the base of the ice remains constant (although different from the uncavitated bed).

For the alternative case of $P^*$ centred around $x_P^* = 4.65$ (dashed blue curve), the formation of a single confined cavity means there is only a single branch of the relationship between $\tau_b^*$ and $1/N^*$. By contrast with the other cases considered above, $\tau_b^*$ now increases without bound as $1/N^*$ increases and, in fact, does so linearly in $1/N^*$ for large $1/N^*$. The reason is simply that a finite cavity size is approached as $N^* \to 0$, and simultaneously a finite $\tau_b$ is approached so that $\tau_b/N$ must increase linearly in $u_b/N$.

We, however, can also view $1/N^* \to \infty$ as the limit of a large velocity $u_b$ rather than the limit of a vanishing effective pressure. Once more, I find linear behaviour analogous to that in Nye (1969) and Kamb (1970) precisely because the confined cavity adopts a constant steady-state shape in the limit of large $u_b$ and therefore has the same effect as a rigid bed in the sense that the base of the ice retains its shape when $u_b$ changes, provided $u_b$ remains large. That shape differs from the case of an isolated cavity discussed above, which explains why the slope of the dashed blue curve for large $1/N^*$ differs from the solid red curve at small $1/N^*$: even though the cavity shape becomes independent of $1/N^*$ in both cases, those cavity shapes and locations differ from one another.

An oddity of the solution with $x_P^* = 4.65$ is that it also exists with negative values $1/N^* < 1/N_{\text{drown}}^* \approx -1.27$ (see inset in Fig. 8); this is not to be interpreted as a valid solution for negative $u_b$ and positive $N$ (which would give negative $N^*$) but arises because although $u_b > 0$ is assumed throughout here, $N^*$ can be negative for $x_P^* = 4.65$ (Fig. 5b).

As a further caveat, note that for a fixed $N$, unbounded $\tau_b$ as shown in Fig. 8 results from the ability to generate arbitrarily large compressive normal stresses on the upstream side of the smaller bed protrusion, balanced by correspondingly negative compressive normal stresses on the downstream side in the hydraulically isolated low-pressure region on the downstream side of the larger protrusion (Fig. 7c2, where $-\sigma_{nn}^*$ is scaled with $1/u_b$, so the actual stress is the pattern shown multiplied by a coefficient proportional to $u_b$). As described in Sect. 2 immediately after Eq. (9), arbitrarily negative normal stresses cannot be generated since a vapour-filled cavity will eventually form, and this should ultimately lead to a bounded basal drag satisfying an amended version of Iken's bound, $\tau_b \leq \max(\partial b/\partial x)p_i$, where $p_i$ is once more overburden. The model here ignores that possibility, effectively treating $p_i$ as infinite for the purposes of bounded basal drag.

## 3.3 More complicated permeable bed portions

The results above were computed either for completely permeable beds or for beds that had permeable sections located at normal stress minima prior to cavity formation. As pointed out, I view these permeable bed portions $P$ as potential proxies for lateral access from a three-dimensional ambient drainage system along an unmodelled part of the ice–bed interface, to one side of the flowline that the model describes. In that case it may make sense for that lateral access to reach the modelled flowline in places where compressive normal stress has local minima. Locating the permeable bed where cavities form at the highest possible values of $N$ is also convenient as it reduces the number of additional parameters that describe the bed in the absence of a more sophisticated three-dimensional model.

In order to investigate the effect of choosing different permeable bed portions $P$, I plot in Fig. 9 the dependence of cavity end-point positions on effective pressure $N^*$ for the same bed geometry (Eq. 10) as before, but for two alternative choices of $P^*$: in panel (a), $P^*$ is the union of the intervals $P_a = (5.890, 6.283)$ and $P_b = (2.316, 2.749)$, while in panel (b), $P^* = P_b$. In both cases, if I start with an uncavitated bed, a cavity first forms around the permeable patch $P_b$ at a critical value $N^* = N_{\text{init}_2}^* \approx 2.00$; this the normal stress $-\sigma_{nn}^*$ at the upstream end $x^* = 2.316$ of the interval $P_b$ when the bed is uncavitated.

Once formed, the cavity immediately has a finite size that extends beyond $P_b$ and is identical to the single-cavity solution shown by the black curve in Fig. 5a. The black curve in Fig. 9a and b traces the growth of the cavity as $N^*$ is lowered below the initiation value $N_{\text{init}_2}^*$ and remains identical to the corresponding portion of the black curve in Fig. 5a, with the cavity expanding past the second smaller bed protrusion at the same critical value $N_{\text{connect}}^*$ as in Fig. 5a.

Conversely, if $N^*$ is increased after initial cavity formation $N_{\text{init}_2}^*$, the single cavity in the lee of the larger bed protrusion remains connected up to a much higher critical pressure $N_{\text{disconnect}_2}^* \approx 6.70$: because the cavity that is established at $N_{\text{init}_2}^*$ extends far beyond the permeably patch, a significant increase in $N^*$ is needed to shrink it to the point at which the connection is lost to the permeable part of the bed patch. This is shown by the red curves in Fig. 9a and b, which remain identical to the black curve in Fig. 5a up to $N_{\text{disconnect}_2}^*$, after which an isolated cavity forms as the downstream contact point moves past the upstream end of the interval $P_b$.

The only difference between the solutions in Fig. 9a and b arises if effective pressure is lowered below the critical value $N_{\text{connect}}^*$, at which the cavity extends past the lower bed protrusion, and subsequently raised to the critical value $N_{\text{disconnect}}^*$ at which the downstream portion of the enlarged cavity becomes separated again by a contact area in Figs. 3a and 5a. Solutions for that situation are shown as blue curves in Fig. 9a and b. In Fig. 9a, the permeable portion $P_a$ in the lee of that smaller bed protrusion keeps the downstream cav-

ity connected up to a critical effective pressure $N^*_{\text{disconnect}_3} \approx$ 4.00. The blue curve in that case remains identical to the solution for a fully permeable bed shown in Fig. 3a up to that point; at $N^*_{\text{disconnect}_3}$, the downstream end of the cavity
attached to the smaller bed protrusion moves past the upstream end of $P_\text{a}$. By contrast, the absence of a downstream permeable interval immediately isolates the downstream cavity when $N^*_{\text{disconnect}}$ is reached in Fig. 9b. The blue curve in Fig. 9b is therefore identical to the solution for an isolated
downstream cavity shown in red in Fig. 5a. This is true until the upstream cavity also disconnects again at a critical pressure $N^*_{\text{disconnect}_4}$ very close to $N^*_{\text{disconnect}_2}$.

These results serve as an illustration of how the placement of drainage access can serve to complicate the computation
of cavity extent. Note, however, that in many cases the solutions shown in Fig. 9 correspond to appropriately spliced-together segments of the simpler solutions of Figs. 3 and 5, with each segment limited by the value of $N^*$ at which a cavity loses connection to $P^*$. The key takeaway is probably
that the location of the permeable patch $P^*$ may make irreversibility under changes in $N^*$ more pronounced: if $P^*$ is not centred around the location of the lowest normal stress for an uncavitated bed, then a fairly low effective pressure $N^*$ may be required to initiate cavity formation (given by
$N^*_{\text{init}_2}$ in Fig. 9), but the cavity that is then formed can remain connected to the drainage system up to much larger effective pressures ($N^*_{\text{disconnect}_2}$ in Fig. 9).

## 3.4  A more complicated bed shape

The results I have found above for the double-humped bed
given by Eq. (10) translate qualitatively to other more complicated bed geometries. Below, I use the following triple-humped periodic bed profile as an illustration.

$$b(x) = h_0 \left\{ \sin\left(\frac{2\pi x}{a}\right) + \frac{1}{2}\left[\cos\left(\frac{4\pi x}{a}\right) \right.\right.$$
$$\left.\left. - \sin\left(\frac{4\pi x}{a}\right)\right] + \sin\left(\frac{8\pi x}{a}\right)\right\} \qquad (15)$$

Figure 10 shows $N^*$ against the location of cavity end
points as in Figs. 3 and 5. I see similar behaviour as for the double-bumped bed: with spatially limited drainage access $P$, cavities can expand to drown bed protrusions in their lee, but not on their upstream side (panel b). In order to drown a lee-side bed protrusion at a positive effective pressure, the
cavity in question needs to be attached to a larger bed protrusion than that being drowned (panel b). That drowning is also irreversible, leaving isolated cavities in place if $N$ is increased again by a sufficient amount (red and blue solution curves in panel b). Where a cavity is attached to a small bed
protrusion upstream of a larger one, it typically remains confined even at small negative effective pressures up to a critical value beyond which force balance is violated and no solution exists (panels c and d).

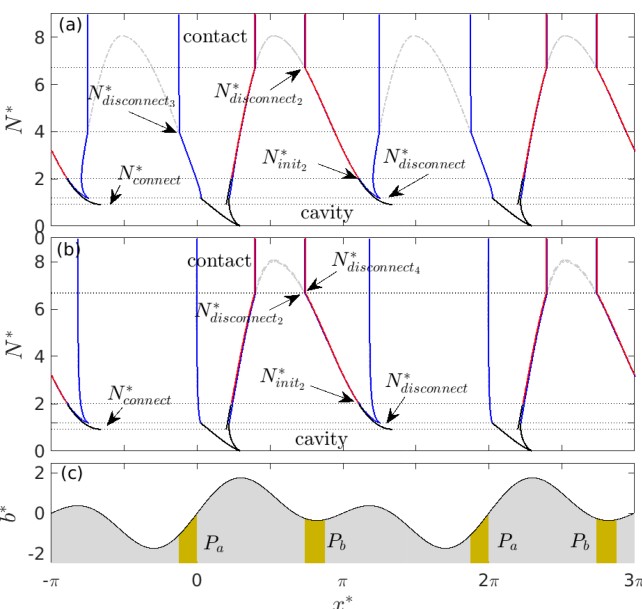

**Figure 9.** The effect of permeable bed patch location, using the same plotting scheme as Fig. 3. Panel **(a)** shows the cavity end-point location against $N^*$ for $P^* = P_\text{a} \cup P_\text{b}$, $P_\text{a} = (5.890, 6.283)$, and $P_\text{b} = (2.316, 2.749)$. Black shows cavity end points obtained when decreasing $N^*$ from the initiation value $N^*_{\text{init}_2}$, and red shows cavity end points obtained when increasing $N^*$ after first establishing a cavity at $N^*_{\text{init}_2}$. Blue (partly obscured by red) shows cavity end points obtained when increasing $N^*$ after first lowering effective pressure below $N^*_{\text{connect}}$. The black curve in Fig. 5a is shown as a dashed grey curve, and the black curve in Fig. 3 is shown as a dot-dashed grey curve. Panel **(b)** shows cavity end points for $P^* = P_\text{b}$. The black curve and red curves are identical to panel **(a)**, and the blue curve is analogous to panel **(a)**, showing the cavity end-point position against $N^*$ if $N^*$ has previously been lowered below $N^*_{\text{connect}}$. The black and red curves in Fig. 5a are shown as dashed and dot-dashed grey curves, the latter almost completely obscured by the blue curve. Panel **(c)** shows the bed and location of $P_\text{a}$ and $P_\text{b}$.

The critical effective pressure at which a cavity extends abruptly across a smaller protrusion in its lee is marked by
50 dotted black lines in Fig. 10b (this is equivalent to $N_{\text{connect}}$ in Fig. 5a, although there are two such critical values in Fig. 10b as there are two smaller bed protrusions in the lee of the largest protrusion). Once the critical effective pressure has been reached and the cavity has extended, contact with the
55 cavity roof can only be re-established by increasing effective pressure to a somewhat different higher effective pressure shown as blue and red dotted lines in Fig. 10b (equivalent to $N_{\text{disconnect}}$ in Fig. 5a; see also the inset in panel b of Fig. 10). At the point of re-contact, an isolated cavity is created be-
60 hind the lee-side bed protrusion. That isolated cavity will reconnect again if the effective pressure is lowered back to the same critical value at which it first formed. In other words, once a cavity has expanded past a lee-side bed protrusion, it

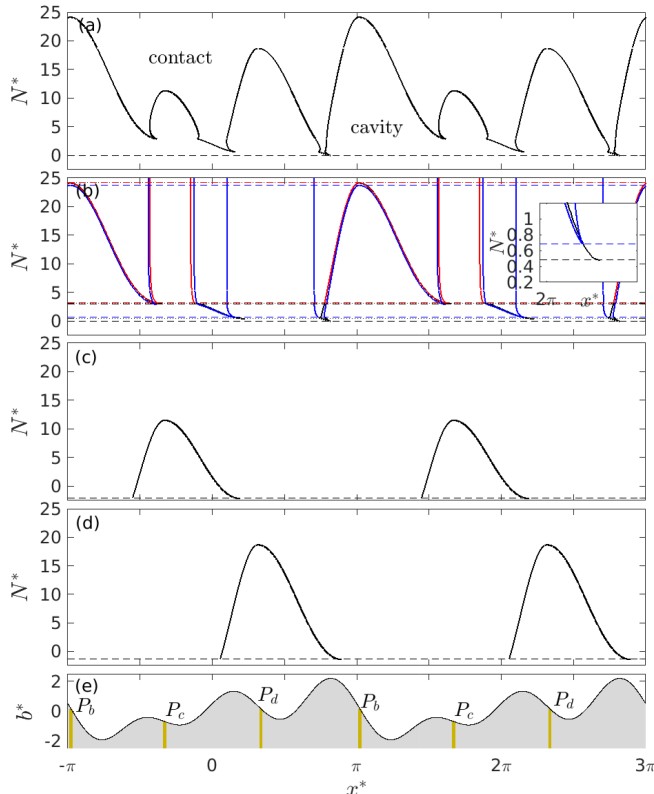

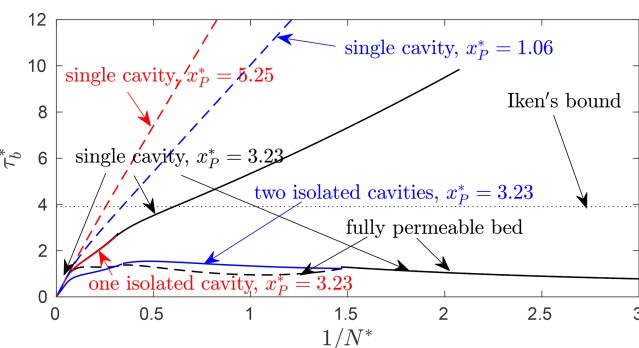

**Figure 11.** Friction law – the CE1 equivalent of Fig. 8 for the bed given by Eq. (15): $\tau_b^*$ against $1/N^*$ for the solutions shown in Fig. 10. Dashed black (partially obscured by solid black as indicated by arrows) corresponds to the permeable bed solution in Fig. 10a. Solid black (multiple segments), red, and blue correspond to the solutions shown in black, red, and blue, respectively, in Fig. 10b. The dashed red curve corresponds to the solution in Fig. 10c and dashed blue to the solution in Fig. 10d. The latter two do extend to some negative values of $1/N^*$ (not shown).

**Figure 10.** Panel **(a)** shows effective pressure $N^*$ against cavity end-point positions for a fully permeable bed with shape given by Eq. (15) as solid black curves. Note that the solution is unique. Panel **(b)** shows the cavity end-point positions for the same bed with a small $P^* = P_b$ centred around $x_P^* = 3.23$ (in the lee of the large bed protrusion; see panel **e**). Black shows the solution for a single cavity initiated around $x_P^*$. Red shows the solution with a single isolated cavity and blue with two isolated cavities. The dashed black line shows values of $N^*$ at which the single cavity expands abruptly, and the dashed red and blue curves show the formation of isolated cavities and the closing of the connected cavity in the presence of one or two isolated cavities. See the inset for details on cavity expansion and formation of an isolated cavity. Panel **(c)** shows the cavity end-point positions for the same bed with a small $P^* = P_c$ centred around $x_P^* = 5.25$ (in the lee of the smallest bed protrusion as shown in panel **e**). The dashed line shows the negative value of $N^*$ at which the cavity no longer remains confined and the ice detaches from the bed. Panel **(d)** shows the cavity end-point positions for the same bed with a small $P^* = P_d$ centred around $x_P^* = 1.03$ (the medium bed protrusion; see panel **e**). Panel **(e)** shows the corresponding bed surface elevation $b^*(x^*)$ defined by Eq. (15) against $x^*$. The beige strips show the permeable areas $P_b$, $P_c$, and $P_d$ used in panels **(b)**–**(d)**, respectively.

will do so again more readily, facilitated by the presence of an isolated cavity behind that protrusion, instead of the original isolated region of ice–bed contact at low normal stress. The disconnection of isolated cavities is reversible, unlike the flooding of low-pressure contact areas.

The friction law for the triple-humped bed (Fig. 11) is more complicated than for the double-humped bed on account of the fact that different numbers of isolated cavities can form, but it again retains similar features: high levels of basal drag $\tau_b^*$ are favoured when smaller lee-side bed protrusions have not been drowned yet or when cavities remain confined in the lee of small bed protrusions. For the latter case, basal drag is again unbounded as $1/N^* \to \infty$. The abrupt expansion of a cavity corresponds to an abrupt drop in drag, as it does in Fig. 8. The lowest levels of basal drag are typically generated for permeable beds and for fully cavitated beds.

One behaviour that differs subtly between the two bed geometries considered here is the dependence of effective pressure in isolated cavities on the effective pressure in connected cavities. For the triple-humped bed (Fig. 12), I see that effective pressure in an isolated cavity directly downstream of the connected cavity increases with forcing effective pressure $N^*$ as in Fig. 6 (with the increase again being rapid when the cavity first forms and then much less than linear in $N^*$). This corresponds to the upward slope of both the blue and red curves near their left-hand starting points, which mark the effective pressures at which the corresponding cavities first become isolated. However, once the larger isolated cavity becomes separated from the connected cavity upstream by an additional isolated cavity in the lee of the smallest bed protrusion, then effective pressure in that larger isolated cavity actually decreases with $N^*$: the blue curve in Fig. 12, representing effective pressure in the isolated cavity in the lee of the second-tallest bed protrusion, actually slopes downwards very slightly underneath the red curve (that is, once there is an isolated cavity in the lee of the smallest bed protrusion,

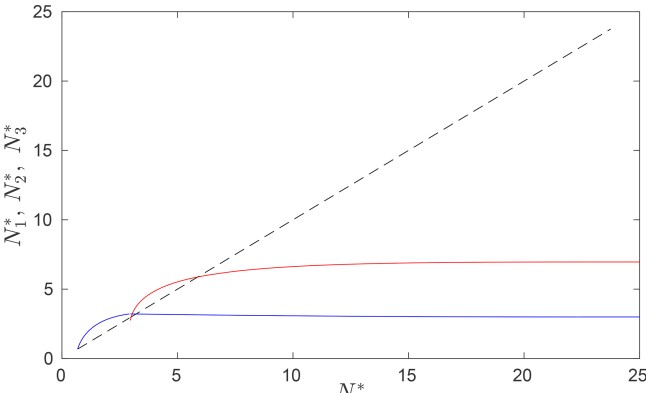

**Figure 12.** Effective pressure in the isolated cavities shown as the blue solution in Fig. 10b against the corresponding effective pressure $N^* = N_2^*$ in the connected cavity (the equivalent of Fig. 6 for the triple-bumped bed given by Eq. 15). Blue shows the effective pressure $N_1^*$ in the isolated cavity around $x^* = 1.03$, and red shows effective pressure $N_3^*$ in the isolated cavity around $x^* = 5.25$, while the black dashed line shows $N_2^* = N^*$ for the range of values of $N^*$ for which the $j = 2$ cavity is open. Note that $N_1^*$ decreases slightly with forcing effective pressure $N^*$ once the second smaller isolated cavity around $x^* = 5.25$ has formed.

which separates the second-tallest from the tallest protrusion as shown in Fig. 10b).

## 4 Discussion

### 4.1 Steady-state subglacial hydrology

The steady-state solutions in Sect. 3 point to three primary insights: first, if the bed is forced by slow changes in drainage system effective pressure $N$ and is therefore always in steady state except during brief transients, then connections to previously uncavitated parts of the bed are made at critical values of $N/u_b$. These critical values depend on the geometry of the bed and on the locations of the parts of the bed that are permeable and therefore intrinsically connected to the ambient drainage system. The model denotes these parts by $P$, and they are indicated by beige colouring throughout the paper.

Second, when such connections occur, they invariably extend the existing cavity in the downstream direction and never upstream. This has major implications for the evolution of connectedness of the bed and for the effective pressures that can be sustained. For cavities that are caused by drainage system access $P$ immediately in the lee of prominent bed bumps, downstream connections occur at positive effective pressures, and smaller bed bumps are submerged by expanding cavities first, as might be expected. If drainage system access $P$ is located in the lee of less prominent bed bumps, then (perhaps counterintuitively) connections are made only once sufficiently negative effective pressures are reached and result in complete ice–bed detachment. Importantly, this im-

plies that sustained negative effective pressures at the glacier bed are possible, as has been inferred from observations (Rada and Schoof, 2018).

Third, once a connection has been made and the lee of a smaller bed protrusion has become submerged, the cavity space on that lee side can subsequently become isolated due to an increase in effective pressure (or decrease in sliding velocity), which causes the cavity roof to be lowered. The critical value for the disconnection between the upstream cavity and newly isolated, cavity however, occurs at a higher critical value $N/u_b$ than the original connection (Fig. 5). Importantly, connection and disconnection become reversible at this point: once the downstream side of a smaller bed bump becomes cavitated, connection and disconnection happen at the same critical value of $N/u_b$. A corollary of this third point is that it is easier to create connections once there are isolated cavities in place, in the sense of that connection happening at a higher value of $N/u_b$ than in the absence of those isolated cavities.

The reader may wonder at this point why one would bother with considering isolated, low-pressure contact areas at the bed at all: since their flooding is irreversible, are they irrelevant, since they will connect sooner or later and henceforth remain flooded, even if they become hydraulically isolated again? The point here is that treating the bed as fully impermeable outside of the region $P$ is likely to be an idealization: in reality, there is almost certainly slow leakage through the "impermeable" bed portions as also envisaged in Hoffman et al. (2016) and Rada and Schoof (2018). If there are lengthy periods outside of the active drainage season (with the latter often occupying a relatively short part of the annual cycle) during which that leakage can drain isolated cavities, then it is possible that the bed starts each season in an uncavitated state. In that case, the expansion of cavities initially confined to locations with access to the drainage system occurs seasonally.

A second point that needs to be addressed here is the limitation imposed by using a two-dimensional domain. True hydraulic connections over distances longer than a single bed wavelength $a$ are clearly only possible in two dimensions if the ice becomes fully detached from the bed, which is clearly not the object of the present study. In reality, hydraulic connections have to be made by connected cavity space that goes around rather than over prominent bed bumps in three dimensions. I anticipate that the results obtained here are still relevant to individual connections between cavities in three dimensions, with those cavities being extended laterally and connecting further downstream or upstream at a lateral offset. Studying these more complicated geometries requires a three-dimensional model (see also Helanow et al., 2020, 2021) that can capture the dynamics of hydraulically isolated cavities and of isolated, uncavitated, low-pressure regions. The model presented in Part 2 is in principle capable of doing that, although in practice I have not been able to run it in a three-dimensional configuration due to compu-

tational constraints: three-dimensional cavity dynamics with hydraulic isolation remain an obvious area of future research.

## 4.2 Steady-state friction law

For a fully permeable bed, the ratio $\tau_{\mathrm{b}}/N$ of basal drag to effective pressure is a single-valued function of the ratio of sliding velocity to effective pressure $u_b/N$ or, more generally, of $u_b/N^n$ for a power-law Glen's law rheology with exponent $n$ (Fowler, 1986; Schoof, 2005; Gagliardini et al., 2007; Helanow et al., 2021). That function behaves roughly as a regularized Coulomb friction law, at least for highly irregular beds (Schoof, 2005; Helanow et al., 2021). By contrast, partial permeability of the bed has a major effect on basal friction: basal drag $\tau_{\mathrm{b}}/N$ now depends not only on $u_b/N$, but also critically on where along the bed the region of drainage access $P$ is located and on whether isolated cavities have previously been formed (Figs. 8 and 11).

The first qualitative difference between a permeable and impermeable bed is that Iken's bound $\tau_{\mathrm{b}} \leq N \max(\partial b/\partial x)$ need not hold for the latter: the derivation of that bound (Schoof, 2005) specifically relies on there being no compressive normal stresses at the bed below the water pressure in the ambient drainage system.

If drainage system access $P$ is located in the lee of one of the smaller bed bumps, then the resulting cavities remain confined and do not lead to widespread ice–bed separation until effective pressure becomes negative as discussed above (see also Figs. 5 and 10). In that case, $\tau_{\mathrm{b}}/N$ increases without bound in $u_b/N$, with the relationship becoming linear at large $u_b/N$ so that $\tau_{\mathrm{b}} \propto u_b$ approximately (see the blue and red dashed curves in Figs. 8 and 11). This result is familiar from Nye–Kamb sliding theory (Nye, 1969; Kamb, 1970) for ice of constant viscosity (as is assumed here) sliding over a rigid bed in the *absence* of cavitation: the confined cavity modifies the shape of the lower boundary of sliding ice, but because cavities do not expand to cover the entire bed as $u_b/N \to \infty$ (as would be the case for a fully permeable bed; see Fig. 3 or 10a), that modification approaches a finite limit for large $u_b/N$, explaining why behaviour analogous to Nye–Kamb sliding is obtained. Importantly, the modification of the lower boundary of the ice depends on the precise location of the confined cavity, and the approximate constant of proportionality relating $\tau_{\mathrm{b}}$ to $u_b$ depends on the location of $P$: this explains, for instance, why there are distinct dashed red and blue lines in Fig. 11.

The most dramatic changes in basal friction occur when $P$ is immediately in the lee of the largest bed bump. In that case, $\tau_{\mathrm{b}}/N$ will increase approximately linearly in $u_b/N$ until the cavity connects with the remainder of the bed (see the solid black curves in Figs. 8 and 11, with the discontinuity that corresponds to the connection point marked as $1/N^*_{\mathrm{connect}}$ in Fig. 8). Iken's bound may be significantly exceeded during that initial increase in $u_b/N$. Once the connection with the remainder of the bed occurs, basal drag $\tau_{\mathrm{b}}/N$ drops dramat-

ically by factors of approximately 3 and 10 in Figs. 8 and 11, respectively. This is not particularly surprising, as the extension of the cavity drowns out much of the previously uncavitated bed topography, forcing the ice to flow over fewer bed obstacles and thereby reducing form drag (that is, drag caused by flow over basal topography).

Once connection has occurred, the friction law mimics the friction law for a fully permeable bed. This remains the case even if $u_b/N$ decreases again to the point where isolated cavities form in the lee of some of the smaller cavities (compare the dashed black curve for a fully permeable bed with the solid red curve for a single isolated cavity in Fig. 8 and with the solid blue curve for isolated cavities in Fig. 11): the smaller obstacles remain drowned once these isolated cavities form, and form drag remains low.

Computation of steady-state friction $\tau_{\mathrm{b}}$ (the dynamic case being even more complicated; see e.g. de Diego et al., 2022 and also Gilbert et al., 2022) therefore requires not only knowledge of $u_b$ and $N$, but also of the prior history of the bed and of hydraulic connections that have been made. This suggests that at least one additional state variable may need to be included in the formulation of steady-state basal friction laws, possibly the cavitation ratio $\theta$ of Thøgersen et al. (2019). The latter is defined as the fraction of the bed that has become cavitated. In fact, the results here suggest that changes in cavitation ratio may have a dominant effect on basal friction: a significant and abrupt increase in cavitation ratio occurs when a cavity extends or "connects" downstream (Figs. 5 and 10), and that increase in cavitation ratio corresponds to an equally abrupt, large drop in basal drag as discussed above.

In fact, a prototype parameterization for the friction laws shown in Figs. 8 and 11 is

$$\tau_{\mathrm{b}} = C(\theta)u_b^{1/n} + \frac{C_0 N u_b^{1/n}}{(\Lambda_0 N^n + u_b)^{1/n}}, \tag{16}$$

where the second term on the right-hand side is the regularized Coulomb friction law of Schoof (2005) and Gagliardini et al. (2007). Here $n$ is the exponent in Glen's law (Paterson, 1994), while $C_0$ and $\Lambda_0$ are constants determined by the geometry of bed roughness. The first term on the right reflects the fact that the friction laws in Figs. 8 and 11 behave as effectively linear relationships when there is an isolated low-pressure region that has not become cavitated yet. In Eq. (16), I propose capturing this behaviour through a linear term in $u_b^{1/n}$ (recall that $n = 1$ in Figs. 8 and 11), with the slope of that linear term depending on the cavitation ratio $\theta$. In order to emulate the behaviour of a fully permeable bed, $C(\theta)$ needs to approach zero for cavitation ratios sufficiently close to unity.

A friction law of this form in turn implies that subglacial drainage models may need to incorporate a description of the evolution of cavitation ratio. As I will show in Part 2, cavitation ratio and mean cavity depth (the variable commonly

used to define cavity geometry in large-scale drainage models) are not simple proxies for each other, implying that the introduction of cavity ratio into friction laws and drainage parameterizations would indeed imply an increase in model complexity.

There is a second complication in the definition of a friction law that deserves to be stressed for an impermeable bed: the quantity that is commonly understood as "effective pressure", overburden minus water pressure at the bed, is not uniquely defined but potentially varies from cavity to cavity. That is, effective pressure varies over length scales that are treated as microscopic in typical subglacial drainage models because water pressure differs between cavities. In the idealized model I use here, I define a unique "ambient drainage system effective pressure" $N$ in the permeable bed portions $P$ and am able to express a friction law in terms of $N$ and $u_b$ (albeit in the form of a multi-valued friction law) as is done in Figs. 8 and 11.

The effective pressure in the connected portion of the drainage system is likely to be the only useful effective pressure that can be defined, as it will in general vary smoothly in space and can therefore be modelled at the large scale, at least in principle. That observation does underline, however, the need to include additional degrees of freedom that capture the degree of cavitation in friction laws, since effective pressure is then meaningless in a part of the bed that is fully hydraulically isolated, with no drainage system access at all: there may still be isolated cavities in that case, and their presence will affect basal friction as discussed above. To compound matters, this situation also significantly complicates any attempts to constrain such a friction law observationally: while effective pressure in a connected drainage system can in principle be measured by borehole access to the bed, the presence and extent of isolated cavities at the bed are much harder to determine.

## 5   Conclusions

Using a simple extension of an existing purely viscous model for steady-state basal cavities in two dimensions, I have shown that uncavitated regions of the bed can persist indefinitely at low normal stress provided there is no drainage pathway along which water can reach them. Such drainage pathways are created under slow changes in forcing effective pressure $N$ when that effective pressure reaches a critical value. The creation of such connections is not reversible by simply raising $N$ back above its critical value, but requires a greater increase in $N$ and leaves behind an isolated cavity. The formation of connections also leads to a significant drop in basal friction that is likewise irreversible, since the isolated cavity that is left behind by a subsequent increase in $N$ significantly reduces contact between the ice and bed even when the hydraulic connection is closed again. To the best of my knowledge, few if any of these phenomena are

included in current large-scale subglacial drainage models or basal friction laws.

The main limitations to the work presented here derive from its assumption of quasi-steady conditions and its restriction to two dimensions. Dynamic cavity connections have significantly richer behaviour than the quasi-steady solution in the present paper suggests and are investigated in detail in a companion paper. Three-dimensional bed topography by contrast remains an open problem and holds the key to a more complete understanding of hydraulic connectivity. Connections at the bed are presumably more likely to occur when bed topography is three-dimensional: in a two-dimensional setting, connectivity along the entire model domain is only possible when ice–bed contact is lost completely, whereas this is not the case in three dimensions. Similarly, contact of the ice roof between two cavities in three dimensions does not necessarily make them disconnected, whereas it does in two dimensions.

## Appendix A: Complex variable solution of the viscous steady-state problem

### A1   Complex variable formulation

The construction in Fowler (1986) and Schoof (2002, pp. 51–54) allows the problem consisting of Eqs. (1), (2), (3), and (5) to be written in the following form: let $z = x + iy$, and find an analytic function $\Omega(z)$ in the complex plane cut along the real axis, satisfying

$$\Omega(z) = \overline{\Omega(\overline{z})}, \tag{A1a}$$

$$-2i\left[\Omega^+(x) - \Omega^-(x)\right] = -N_j \qquad \text{for } x \in C_j, \tag{A1b}$$

$$\Omega^+(x) + \Omega^-(x) = \eta u_b b''(x) \qquad \text{for } x \in C', \tag{A1c}$$

$$\Omega(z) \to 0 \qquad \text{as } \Im(z) \to \pm\infty, \tag{A1d}$$

where a prime indicates differentiation (in this case, with respect to $x$), an overbar signifies complex conjugation, and superscripts $+$ and $-$ denote limits taken from above and below the real axis. The constraints (7) and (9) become

$$h_C(x) > b(x) \qquad \text{for } x \in C, \text{ where} \tag{A1e}$$

$$\Omega^+(x) + \Omega^-(x) = \eta u_b h_C''(x)$$

$$\text{and } h_C(b_j) = b(b_j), \tag{A1f}$$

$$h_C(c_j) = b(c_j); \tag{A1g}$$

$$-2i\left[\Omega^+(x) - \Omega^-(x)\right] > N_j \qquad \text{for } b_j - \delta < x < b_j \tag{A1h}$$

$$\text{and } c_j < x < c_j + \delta \tag{A1i}$$

and some finite $\delta$.

Let $\zeta = \exp(i2\pi z/a)$ and $\xi = \exp(i2\pi x/a)$. The assumed periodicity of the solution ensures that $\Omega(z)$ can be mapped one to one to $G(\zeta) = \Omega(z)$, and similarly $b_2(\xi) = b''$ and $h_{2C}(\xi) = b_C''$ TS4 are one-to-one mappings. The func-

tions $G$, $b_2$, and $h_{2C}$ satisfy

$$G(\zeta) = \overline{G(1/\bar{\zeta})}, \tag{A2a}$$

$$G(\infty) = 0, \tag{A2b}$$

$$-2i\left[G^+(\xi) - G^-(\xi)\right] = -N_j \qquad \text{for } \xi \in \Gamma_j, \tag{A2c}$$

$$G^+(\xi) + G^-(\xi) = \eta u_b b_2(\xi) \qquad \text{for } \xi \in \Gamma', \tag{A2d}$$

$$G^+(\xi) + G^-(\xi) = \eta u_b h_{2C}(\xi) \qquad \text{for } \xi \in \Gamma, \tag{A2e}$$

where $\Gamma_j$, $\Gamma$, and $\Gamma'$ are $C_j$, $C$, and $C'$ mapped into the complex $\zeta$ plane (where they become subsets of the unit circle), and $+$ and $-$ now indicate limits taken from within and without the unit circle in the $\zeta$ plane.

The solution method followed here is that of Schoof (2002, 2005), slightly modified to account for cavities at different effective pressures. I outline the procedure in full below, adding detail omitted in the original account by Schoof (2002, 2005).

## A2  Cavity roof re-contact constraints

As in Schoof (2005), it is possible to conclude that the cavity roof must disconnect and reattach tangentially and that it suffices to impose this on $n-1$ of $n$ cavities since any valid solution to Eq. (A2) ensures that re-contact is then also tangential for the $n$th cavity. Consider the integral

$$I = \int_0^a \Omega^+(x) + \Omega^-(x)\mathrm{d}x$$

$$= \eta u_b \left\{ \sum_{j=1}^n \left[h_C'(c_j) - b'(c_j)\right] \right.$$

$$\left. - \sum_{j=1}^n \left[h_C'(b_j) - b'(b_j)\right] \right\}, \tag{A3}$$

where I have used Eqs. (A1c) and (A1g). Enforcing the contact condition (A1e) combined with the constraint that $h_C(b_j) = b(b_j)$, $h_C(c_j) = b(c_j)$ implies that $h_C'(c_j) \leq b'(c_j)$, $h_C'(b_j) \geq b'(b_j)$, and hence $I \leq 0$. On the other hand, transforming to the $\zeta$ plane,

$$I = \frac{a}{2\pi i} \int_{\Gamma \cup \Gamma'} \frac{G^+(\xi) + G^-(\xi)}{\xi}\mathrm{d}\xi = 0 \tag{A4}$$

on account of Cauchy's theorem, since $\Gamma \cup \Gamma'$ is the unit circle and therefore a closed contour, and $G(0) = G(\infty) = 0$. $I = 0$ in turn implies that the cavity roof detaches and re-contacts tangentially, so

$$h_C'(c_j) = b'(c_j), \qquad h_C'(b_j) = b'(b_j) \tag{A5}$$

for $j = 1, \dots, n$.

In fact, tangential cavity roof detachment and re-contact are required not only by Eq. (A4), but also by the original

construction of the model in Eq. (A1), which requires differentiation of the original normal velocity condition $v = u_b b'$ or $v = u_b h_C'$ (Schoof, 2002, p. 44); recovery of the original boundary condition in terms of antiderivatives of $\Omega$ confirms that no discontinuity between $h_C'$ and $b'$ can appear if $\Omega$ is sectionally holomorphic in the sense of Muskhelishvili (1992) (meaning it gives rise to an integrable stress field).

The point here is really to account for the independent number of constraints on the solution that arise from the tangential re-contact. In integrating Eq. (A1g) (or Eq. A2e), the relevant continuity constraints can always be imposed on one cavity end point (say, the upstream end), and integration forward to the other cavity end point then creates a constraint on the solution. Thus, integrating once, I obtain $n$ equations of the form

$$b'(c_j) = b'(b_j) + \int_{b_j}^{c_j} \Omega^+(x) + \Omega^-(x)\mathrm{d}x, \tag{A6}$$

where $\Omega^\pm(x) = G^\pm(\exp(i2\pi x/a))$. Integrating twice, I obtain another $n$ constraints

$$b(c_j) = b(b_j) + b'(b_j)(c_j - b_j)$$

$$+ \int_{b_j}^{c_j} (c_j - x)\left[\Omega^+(x) + \Omega^-(x)\right]\mathrm{d}x. \tag{A7}$$

Note, however, that one of the $n$ constraints (A6) is redundant for a valid solution $G$ satisfying $G(0) = G(\infty) = 0$, since this ensures that $I = 0$ and the remaining equation of the form in Eq. (A6) is automatically satisfied.

## A3  Solution

Armed with this result, I can again follow the same solution procedure as in Schoof (2002). $G$ can be written in the form (Muskhelishvili, 1992)

$$G(\zeta) = \frac{1}{2\pi i}\left[\sum_j \int_{\Gamma_j} \frac{-iN_j/2}{\chi^+(\xi)(\xi - \zeta)}\mathrm{d}\xi \right.$$

$$\left. + \int_{\Gamma'} \frac{\eta u_b b_2(\xi)}{\chi^+(\xi)(\xi - \zeta)}\mathrm{d}\xi + P(\zeta)\right]\chi(\zeta), \tag{A8}$$

where $P$ is a polynomial and $\chi$ is a Plemelj function, holomorphic in the complex plane cut along $\Gamma'$, on which it satisfies $\chi^+(\xi) + \chi^-(\xi) = 0$. There are multiple choices of $\chi$ that give rise to a sectionally holomorphic solution $G$, differing in the number and location of singularities at the cavity end points $x = b_j$ and $x = c_j$. As in Fowler (1986) and Schoof (2005), I default to the choice

$$\chi(\zeta) = \prod_{j=1}^n \left(\frac{\zeta - \xi_{b_j}}{\zeta - \xi_{c_j}}\right)^{1/2}, \tag{A9}$$

behaving as $\chi \to 1$ as $\zeta \to \infty$, with $\xi_{b_j} = \exp(i2\pi b_j/a)$ and $\xi_{c_j} = \exp(i2\pi c_j/a)$. This choice of $\chi$ generally places a stress singularity at cavity re-contact points $x = c_j$ but ensures that stress is continuous at detachment points $x = b_j$. That choice is not arbitrary: in Sect. A5 I confirm that stress at $x = b_j$ must be continuous in order to simultaneously satisfy Eqs. (A1e) and (A1i) and that, in general, the stress field at $x = c_j$ will be singular when the same constraints are satisfied locally near the re-contact point.

In order for $G$ to satisfy $G(\infty) = 0$ with $\chi$ given by Eq. (A9), $P \equiv 0$ is necessary and sufficient. The remaining constraint on $G$ is that $G(\zeta) = \overline{G(1/\bar{\zeta})}$; when the latter is satisfied, $G(0) = G(\infty) = 0$ follows automatically. Again as in Schoof (2005, p. 618), it is possible to show that

$$\overline{\chi^+(\xi)} = \begin{cases} -\chi^+(\xi)/\chi(0) & \text{on } \Gamma', \\ \chi^+(\xi)/\chi(0) & \text{on } \Gamma, \end{cases}$$

$$\bar{\xi} = \frac{1}{\xi}, \qquad \overline{\mathrm{d}\xi} = -\frac{1}{\xi^2}\mathrm{d}\xi. \tag{A10}$$

Using these, it follows that

$$\overline{G(1/\bar{\zeta})} = G(\zeta) - \frac{1}{2\pi i}\left[\int_{\Gamma'} \frac{\eta u_b b_2(\xi)}{\chi^+(\xi)\xi}\mathrm{d}\xi \right.$$
$$\left. + \sum_{j=1}^{n} \int_{\Gamma_j} \frac{-iN_j/2}{\chi^+(\xi)\xi}\mathrm{d}\xi\right]\chi(\zeta), \tag{A11}$$

and the required constraint is to set the term in square brackets to zero,

$$J := \int_{\Gamma'} \frac{\eta u_b b_2(\xi)}{\chi^+(\xi)\xi}\mathrm{d}\xi + \sum_{j=1}^{n} \int_{\Gamma_j} \frac{-iN_j/2}{\chi^+(\xi)\xi}\mathrm{d}\xi = 0. \tag{A12}$$

Suppose that the $N_j$ is prescribed. With $P \equiv 0$, the solution $G$ in Eq. (A8) contains $2n$ unknown parameters in the form of the cavity end-point locations $\xi_{b_j}$ and $\xi_{c_j}$. Assuming that $G(\zeta) = \overline{G(1/\bar{\zeta})}$ so $G^+ + G^-$ is real, I have $2n - 1$ real constraints through Eqs. (A6) and (A7). This leaves a single real constraint to close the system, and it therefore remains to show that Eq. (A12) constitutes that single real equation. Taking the complex conjugate of the left-hand side of Eq. (A12) and using Eq. (A10), it is possible to show that $\bar{J} = \chi(0)J$. Since $\chi(0) = \exp[i\pi \sum_{j=1}^{n}(b_j - c_{j-1})/a]$ (Schoof, 2002, p. 98) and $0 < \sum_{j=1}^{n}(b_j - c_{j-1})/a < 1$, it follows that the real and imaginary parts of $\chi(0)$ are non-zero, and hence $\Re(J) = 0$ implies $\Im(J) = 0$ and vice versa. Equation (A12) therefore constitutes a single real constraint, and together with Eqs. (A6) and (A7) I have $2n$ real constraints to determine the $2n$ cavity end points. Prescribing cavity volume $V_j$ rather than effective pressure $N_j$ does not lead to further complications since putting $V_j = \int_{b_j}^{C_j} h_c(x)\mathrm{d}x$ simply

adds the required additional constraint to determine the corresponding $N_j$. The implementation of Eqs. (A12), (A6), and (A7) (combined with additional constraints on $N_j$ when cavity volume is prescribed) follows the same numerical method as in Schoof (2002).

## A4 Arc length continuation

In practice, I introduce the smallest new cavity possible when the inequality (6) is violated somewhere on $P$ (note that this is generally simple to do when $P$ is a small region around the location $x_P$ where normal stress has a local minimum in the absence of cavitation). I then use an arc length continuation to solve the system of equations in Eqs. (A6), (A7), and (A12) while decreasing the effective pressure $N$, forcing cavity end points to change continuously where they can.

Neighbouring cavities $j$ and $j+1$ can merge when $c_j = b_{j+1}$ for some critical value of $N$, in which case I simply deleted $c_j$ and $b_{j+1}$ and created a single enlarged cavity with end points $b_j$ and $c_{j+1}$. Abrupt enlargement of cavities into a previously uncavitated low-pressure region occurs when the solution computed by arc length continuation violates the local constraints in Eqs. (7) and (9) near a cavity end point. This generally corresponds to a fold bifurcation along the solution curve (plotting cavity end-point locations against $N$). $N$ begins to increase again along the curve at such a fold, signalling that the actual solution under a further decrease in $N$ is not continuous. I use arc length continuation to extend the solution further until I reach another solution with the value of $N$ at the fold bifurcation, but for which the inequalities (7) and (9) are satisfied. I treat that solution as representing the enlarged cavity that results from decreasing $N$ past the fold bifurcation and discard solutions computed by arc length continuation that do violate the inequalities in Eqs. (7) and (9).

In order to capture the effect of cavity isolation, I compute solutions by arc length continuation under increases in $N$, checking whether inequalities in Eqs. (7) and (9) are satisfied. An isolated cavity forms when the cavity roof contact constraint (7) is violated in the interior of a cavity. In that case, I introduce new contact points where re-contact occurs and check whether either of the two new cavities created in the process no longer straddles $P$. Such a cavity is then isolated. I compute its volume and restart a computation by arc length continuation while imposing Eq. (8) for that cavity.

## A5 Cavity end-point singularities revisited

Here I show that continuous stress at cavity detachment points and a stress singularity at reattachment points are natural consequences of the inequality constraints (A1e) and (A1i). I use the complex variable formulation deployed above, but note that the same result could be obtained by looking for a stream function solution of the Stokes flow

problem (Eq. 1) in terms of local polar coordinates centred at $x = b_j$ or $x = c_j$ (see e.g. Fontelos and Muñoz, 2007).

Consider the original Hilbert problem (Eq. A1) locally, in a neighbourhood of a cavity end point $z = b_j$ or $z = c_j$. Consider first the detachment point $b_j$, and let

$$
\Omega(z) =
\begin{cases}
-i N_j/4 + \eta u_b b''(b_j) + F(z) & \text{for } \Im(z) > 0 \\
i N_j/4 + \eta u_b b''(b_j) + F(z) & \text{for } \Im(z) < 0.
\end{cases}
\tag{A13}
$$

Then, in some sufficiently small open disc $D$ around $z = b_j$, $F(z) = \overline{F(\overline{z})}$ is holomorphic with a branch cut $L$ along the intersection of $D$ with the half-line $L_0$ given by $y = 0$, $x < b_j$. On that branch cut

$$
F^+(x) + F^-(x) = u_b[b''(x) - b''(b_j)].
\tag{A14}
$$

Assuming again that $F$ is sectionally holomorphic in the disc to ensure integrable stresses, the solutions in the disc take the form (Muskhelishvili, 1992)

$$
F(z) =
\left\{ \Phi(z) + \frac{1}{2\pi i} \int_L \frac{u_b\left[b''(x) - b''(b_j)\right]}{\chi^+(x)(x - z)} \, dx \right\} \chi(z).
\tag{A15}
$$

Here, $\chi(z) = (z - b_j)^{-1/2}$ is analytic in the plane cut along $L_0$, behaving as $\chi(x) = 1/\sqrt{x - b_j}$ for $x > b_j$ along the real axis, and $\Phi$ is holomorphic in $D$. Assume $b''$ is continuously differentiable. Then the limiting values as $y \to 0$ of the integral in the curly brackets behave as a constant plus a term of $O(|z - b_j|^{3/2} \log |z - b_j|)$ (by a straightforward adaptation of the derivation in Muskhelishvili, 1992, pp. 45–49), while the analytic function $\Phi$ can be expanded as a Taylor series around $z = b_j$ as $\Phi(z) = a_0 + a_1(z - b_j) + O(|z - b_j|^2)$. To ensure that $F(z) = \overline{F(\overline{z})}$, $a_0$ and $a_1$ must be real. To a quadratic error in $(z - b_j)$, I simply have $F(z) \sim \left[a_0 + a_1(z - b_j)\right](z - b_j)^{1/2}$, and I can evaluate

$$
\eta u_b h_C'' = \Omega^+(x) + \Omega^-(x)
$$
$$
\sim \eta u_b b''(b_j) + 2[a_0 + a_1(x - b_j)]/\sqrt{x - b_j}
$$
$$
\text{for } x > b_j,
\tag{A16}
$$

$$
p - 2\eta \frac{\partial v}{\partial x} = -2i\left[\Omega^+(x) - \Omega^-(x)\right]
$$
$$
\sim -N_j - 4\left[a_0 + a_1(x - b_j)\right]/\sqrt{b_j - x} \geq -N_j
$$
$$
\text{for } x < b_j.
\tag{A17}
$$

Since $h_C' = b'$ from Sect. A2, it follows that I must have $h_C''(x) \geq b''(b_j)$ for $x > b_j$ in order to ensure that $h_C > b$ inside the cavity. It follows that $a_0 \geq 0$, with $a_1 \geq 0$ if $a_0 = 0$. In order for the normal stress constraint (A17) to be satisfied, ensuring the cavity remains sealed (by considering a local solution, I can dispense with the machinery of requiring a constraint only over a finite region of size $\delta$ as in Eq. A1i), I

see that I must have $a_0 \leq 0$ and $a_1 \geq 0$ if $a_0 = 0$. The only way that both of these constraints can be satisfied is that $a_0 = 0$ and $a_1 \geq 0$. This immediately ensures that normal stress $p - 2\eta \partial v/\partial x \sim -N_j + a_1\sqrt{b_j - x}$ is non-singular at the detachment point.

The same approach can be used near a re-contact point, but with different conclusions. Replacing $b_j$ by $c_j$, I can still define $F$ inside a small open disc $D$ centred on $z = c_j$ through Eq. (A13). $F$ still satisfies Eq. (A14), but now on the intersection $L_0$ of $D$ with the half-line $L = \{(x, 0) : x > c_j\}$. Similarly, $\chi(z) = (z - c_j)^{-1/2}$ is holomorphic in the plane cut along $L_0$, behaving as $1/\sqrt{x - c_j}$ when the branch cut is approached from the upper half-plane. $F(z) = \overline{F(\overline{z})}$ now requires that I write $\Phi(z) = i\left[a_0 + a_1(z - c_j)\right] + O(|z - c_j|^2)$ with $a_0$ and $a_1$ real. The equivalent of Eq. (A16) and inequality (A17) becomes

$$
\eta u_b h_C'' = \Omega^+(x) + \Omega^-(x)
$$
$$
\sim \eta u_b b''(c_j) + 2[a_0 + a_1(x - c_j)]/\sqrt{c_j - x}
$$
$$
\text{for } x < c_j,
\tag{A18}
$$

$$
p - 2\eta \frac{\partial v}{\partial x} = -2i\left[\Omega^+(x) - \Omega^-(x)\right]
$$
$$
\sim -N_j + 4\left[a_0 + a_1(x - c_j)\right]/\sqrt{x - c_j} \geq -N_j
$$
$$
\text{for } x > c_j.
\tag{A19}
$$

With $h_C' = b'$ at $x = c_j$, I must still have $h_C'' \geq b''$ for $x < c_j$ to ensure that $h_C > b$, and hence $a_0 \geq 0$ with $a_1 \leq 0$ if $a_0 = 0$ from Eq. (A18). Satisfying the normal stress condition (A19) requires that $a_0 \geq 0$, with $a_1 \geq 0$ if $a_0 = 0$. In general I therefore expect a solution with $a_0 > 0$ and a singular normal stress of the form $p - 2\eta \partial v/\partial x \sim 4a_0/\sqrt{x - c_j}$.

*Code availability.* The code used is available from the author on request.

*Data availability.* No data sets were used in this article. TS5

*Competing interests.* The author has declared that there are no competing interests.

*Acknowledgements.* This work was supported by NSERC Discovery Grant RGPIN-2018-04665. I would like to acknowledge the helpful comments of two anonymous referees, which improved the paper significantly.

*Financial support.* This research has been supported by the Natural Sciences and Engineering Research Council of Canada (grant no. RGPIN-2018-04665).

*Review statement.* This paper was edited by Nanna Bjørnholt Karlsson and reviewed by two anonymous referees.

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

**Remarks from the language copy-editor**

CE1    Please note the slight edits (two colons can't grammatically be used one after the other in this way).

**Remarks from the typesetter**

TS1    Please give an explanation for why this needs to be changed. We have to ask the handling editor for approval. Thanks.

TS2    Please give an explanation for why this needs to be changed. We have to ask the handling editor for approval. Thanks.

TS3    Please give an explanation for why this needs to be changed. We have to ask the handling editor for approval. Thanks.

TS4    Please give an explanation for why this needs to be changed. We have to ask the handling editor for approval. Thanks.

TS5    Please note that the data availability section is mandatory. Please confirm once more that no underlying data has been used and that this sentence is correct.

TS6    Will be updated by Copernicus Production Office.