# Peer review of "The evolution of isolated cavities and hydraulic connection at the glacier bed. Part 1: steady states and friction laws"

_EGUsphere, 2022_

## Author Comment (AC1)

**The evolution of isolated cavities and hydraulic connection at the glacier bed. Part 1: steady states and friction laws**

Christian Schoof

May 21, 2023

**1 Referee #1**

I'd like to thank referee # 1 for their attention to detail, which is (sadly) becoming unusual.

**Referee:** 1. In the model setup and introduction, the explanation of 'access' (to ambient drainage) and 'permeable patch' is unclear. The current combination of text and Figure 1 makes these ideas confusing and harder to understand than necessary, through to page 4 (past Figure 1). Only quite far in Section 2 do I know for sure from the mathematical descriptions how drainage connections are set up in your model and what they mean. You should stabilise the terminology and clarify /elaborate on the meaning of various terms and phrase ideas more carefully. Here are some of the key issues:

p2, lines 26-28: "Access to ambient drainage system is defined through a permeable bed patch P on which effective pressure N is prescribed; elsewhere, effective pressure is defined through... connectedness to patch P, or through...". I found this passage to be cryptic. The indirect phrasing "is defined through" (used twice) causes vagueness. What $N$ refers to isn't clear, except it is an effective pressure. What "access" means isn't clear, nor how its meaning differs from "connection/connectedness". In "permeable bed patch $P$", I find the word "permeable" to be distractive; I guess it is used to relate P to the context of the permeable part of the wider/ambient subglacial system, but currently I sense that possibly the permeability of P will be modelled or quantified (but it isn't). Consider using direct phrasing such as "$P$ locates the connection, where $N$ is fixed as...".

**Response:** I have kept "permeable" as the operative phrase here (rather than "access"), because that is what I mean: the bed (as in, the substrate below the ice and any cavity space) *is permeable* in the region $P$, in the usual sense of a porous medium. That is to say, that water can flow through the bed in $P$ (not just along the ice-bed interface through available cavity space), and this is to be contrasted with an impermeable bed where water cannot flow: the part of the bed that is not $P$ is completely impermeable.

The point is that existing cavity formation models assume the bed is not just permeable, but highly permeable, allowing water to flow to any part of the ice-bed interface with negligible pressure gradients, so water that fills cavity space there is at the same pressure as the ambient drainage system. I have tried to clarify this. The original third paragraph of the introduction stated

*In process-scale models, hydraulic connectedness typically occurs through the bed itself: the bed is highly permeable, offering ready access to water sourced from an ambient drainage system at some given water pressure. That water will force its way between ice and bed as soon as compressive normal stress in the ice drops to the level of the water pressure, causing a cavity to form ...*

I've re-worded this slightly as

*In process-scale models, hydraulic connectedness typically occurs through the bed itself: the bed is highly permeable. Water sourced from an ambient drainage system at some given water pressure can force its way between ice and bed as soon as compressive normal stress at the base of the ice drops to the water pressure in the ambient drainage system, causing a cavity to form. . .*

I assume that $P$ retains that the same highly permeable properties assumed for the whole bed by existing models for cavitation. I have changed the passage highlighted in the review to the following: *In the present work, I have used amodified mathematical model for cavity formation to explore the physics involved. The basic physics of ice flow over an undulating bed, allowing for the possibility of ice-bed separation as water forces its way between the two, is the same as in existing models for subglacial cavity formation. However, only a pre-defined, highly permeable part of the bed, denoted by $P$ is assumed to be directly connected to the ambient drainage system: as in the existing models of de Diego et al (2021, 2022), Gagliardini et al (2007), Helanow et al (2020,2021), Schoof (2005), Stubblefield et al (2021), water is assumed to force its way between ice and bed if interfacial normal stress on $P$ drops to the value of the water pressure in the ambient drainage system. The remainder of the bed is assumed to be completely impermeable. Water can access these other parts of the bed interface (outside of $P$) only if there is a hydraulic connection to $P$ along the ice-bed interface. Moreover, if water has previously accessed some impermeably part of the bed and the hydraulic connection has subsequently been closed, then an isolated cavity is formed. The water pressure in that isolated cavity can differ from the water pressure in the ambient drainage system, but the volume of the cavity will remain fixed.*

Again, I have used "permeable" because that is what we assume in cavity formation models: water flows *through* the bed and emerges at the ice-bed interface where a cavity forms, and when I say permable, I mean highly permeable, as the paragraph hopefully makes clear. I subsequently stop stressing the "highly" part of "highly permeable", on the assumption that this is implicit once I have defined the model mathematically.

I have also tried to make the next paragraph more explicit: *The model comes in two flavours: first, a two-dimensional, purely viscous flow model for the ice, assumes the cavity roof is in steady state, and water pressure in each separate cavity is spatially uniform. Where a cavity overlaps with the permeable part $P$ of the bed, water pressure equals that in the ambient drainage system, while water pressure in isolated cavities is dictated by their volume. Ssecond, a more general dynamic model assumes viscoleastic ice flow and explicitly considers how water is redistributed within the cavities by water pressure gradients, in a manner analogous to hydrofracture models for pre-existing cracks. The hydraulic conductivity that controls water flow is large within cavities (ensuring rapid equilibration) but vanishes when the ice-bed gap is zero, thereby allowing the model to capture the formation of isolated cavities and of isolated but uncavitated low-pressure regions in a dynamic framework.*

I do say later in the paper that I'm trying to mimic lateral access from an ambient drainage system — imagine a channel off to one side and a sequence of connected cavities linking the flowline under consideration to that channel, for instance. Clearly, a two-dimensional model cannot achieve that, but it's likely to be the reality that I'm trying to get closer to in terms of modelling. *However*, a clean statement of the model really has to talk about water access through the bed, because that's how the existing models on which I build work (even if implicitly): they do not concern themselves at all with the topology of cavities, even in three dimensions. As soon as compressive normal stress drops to the level of the ambient water pressure, then water flows to fill a cavity at that location, and this must generally happen through the bed, so yes, vertically. I'm modifying those models here to at least be able to say something useful about hydraulically isolated portions of the bed.

For what it's worth, I'm not sure I can do better than the above while still representing what I

mean to say; in the end, the more unambiguous language of mathematics will have to speak for itself.

**Referee:** Across p2 to p4, P also switches between "patch", "location", "portion", "access". For example, "the location $P$ of ambient drainage system access" on p3 (line 7) is difficult to understand at that stage of the manuscript. Overall the descriptions of this topic towards the end of Section 1 don't communicate the picture/setup well as a foundation for Section 2. The mix of terms also conflicts with Fig. 1, which uses what looks like a vertical line to portray P (yet the caption refers to 'portion').

Fig 1 and its caption: In specifying part P of the bed interface, I think you have at least lateral connection in mind, as described mid-page on p5. Up to page 4, the concept of a vertical connection doesn't seem to be an emphasis. The vertical line therefore confused me: it doesn't indicate a horizontal extent, and I didn't understand how it portrayed "patch" or "portion" as it is so thin. The caption says "beige . . . permeable portions (plural)" but this refers to figures throughout the paper and doesn't specifically explain the current figure. I think you need to rework the caption, adding text to (i) clarify what P means and how the figure portrays it (lines 2-3 currently do not do a good job), and (ii) say explicitly that Cavity 1 is connected because it overlaps with P (/is intercepted by P), explaining also why Cavity 2 is unconnected. [Explanation (ii) is given in the text later, above equation (8), but I think you need it also in the figure caption.]

**Response** I can't promise that I've dealt with this to satisfaction, but in addition to the changes described above, I've altered are the following last paragraph of the intro to

*. . . as well as on the location of the permeable part $P$ of the bed directly connected to the ambient drainage system, . . .*

replaced "portion" with "part" in the figure caption, and changed the last sentence in the figure caption to

*Cavity $j = 2$ here is an isolated cavity, with fixed volume $V_2$, while cavity $j = 1$ is connected as it overlaps with $P$.*

I confess I did not change the figure caption beyond that; in my understanding from previous dealings with scientific editors, figure captions are supposed to contain only the absolutely necessary information, so I've tried to let the main text speak for itself. I hope I have defined $P$ more clearly in the introduction, which precedes the figure. (The figure being a float, I don't really have control over the precise location, but it's intended to be placed in section 2 and is meant to be read in conjunction with the main running text?) In addition, I'm not sure why a narrow region $P$ in the figure is problematic (as oppsed to a wider region); I've used that in figure 1 mostly because it's what most (but no longer all) of the relevant computations later in the paper also use. Hopefully the change of wording from "portion" to "part" will cover that? I have made $P$ into a wider strip in the schematic just in case.

I've also made a handful of tweaks to the running text of section 2 to try to make the use of $P$ more consistent. For instance, in the paragraph leading up to equation (8), I've added the note *(P is a part of the bed, but specified here only in terms of the horizontal coordinates of points in $P$ at the ice-bed interface, since no depth-dependent physics in the bed is resolved by the model.)* and later, two paragraphs after equation (9), I've reworded the second sentence as *Strictly speaking, water here is assumed to flow* through *the permeably bed in $P$ in order to access connected cavities, but $P$ can also be thought of as locations where an ambient drainage system is able to access the being modelled laterally along the ice-bed interface, with the lateral dimension remaining unresolved.*

**Referee:** 2. Choice of $x_P$:

Although you give motivation for placing P around "where cavities first form [in the fully permeable case]" (p5, line 20) — close to the middle/steepest point on lee sides, this choice is restrictive. What

if the connection occurs elsewhere on a lee side, away from that location, or on a stoss side? The analysis would be more complete if the former (lee-side) case at least is discussed properly, preferably aided by a suitable experiment to show the resulting behaviour.

You choose P to be narrow. What happens if it is wide/wider zone (even if you continue to assume only one such zone in (0, a))? Brief discussion of this scenario would be useful.

**Response:** I have added a new section 3.3 on "More complicated permeable portions". I will not repeat what this says word for word here, but suffice it to say that the section uses two illustrative examples to show when the solution using the simple choices of $x_P$ in section 3.1 can and cannot be used to predict the behaviour for more complicated $P$. There is, of course, no "general solution" here, and every specific case would need to be computed on its own merits.

**Referee** 3. As reported clearly in the manuscript, there can be different solutions (cavity configurations) for the same $N^*$ depending on evolution history (in the quasi-steady sense), depending on whether an unconnected lee zone has yet been flooded — e.g. at $N^*$ slightly above $N^*_{disconnect}$ in Fig. 5a (and within the corresponding sequence in Fig. 4). Does the solution method include or require a specific device or algorithm that tracks an aspect of the history to reach the different solutions, e.g. some iteration that uses the last configuration as initial guess? I might have missed it in the descriptions of p5 and the appendix, or perhaps it isn't necessary and I misunderstand the solution method in (A3), or relevant details exist in the referenced literature. Clarification about this in the manuscript would be welcome.

**Response:** The last paragraph of section had been intended to cover this, but was undoubtedly short on detail. In order not to break the flow of the paper for less technically-minded readers, I have adapted the existing text there slightly, and provided detail in a new appendix A.4:

*In the next subsection, I consider a system of cavities that is in quasi-equilibrium, forced by a very slowly changing effective pressure N in the ambient drainage system. I also assume that the bed starts with no cavities. The latter initially form around the permeable parts P of the bed when N is made sufficiently small. The cavities at first remain trapped between prominent protrusions, but can drown these bed protrusions abruptly when N is decreased to some critical values; I describe the method by which I compute the enlarged cavity in detail in appendix A.4. If N is increased again, the extended cavity roof can then make contact again with the drowned bed protrusion, thereby (in two dimensions) sealing the lee side of that protrusion and forming an isolated cavity. The volume of that isolated cavity is dictated by cavity size at the point where the cavity roof re-contacts, making the solution unique for a sequence of slow changes in N. Again appendix A.4 provides further detail.*

Appendix A.4 states the following:

*In practice, I introduce the smallest new cavity possible when the inequality (6) is violated somewhere on P (note that this is generally simple to do when P is a small region around the location $x_P$ where normal stress has a local minimum in hte absence of cavitation). I then use an arc length continuation to solve the system of equations (A6), (A7) and (A12) while decreasing the effective pressure N, forcing cavity end points to change continuously where they can.*

*Neighbouring cavities j and j + 1 can merge when $c_j = b_{j+1}$ for some critical value of N, in which case I simply deleted $c_j$ and $b_{j+1}$ and create a single enlarged cavity with end points $b_j$ and $c_{j+1}$ Abrupt enlargement of cavities into a previously uncavitated low-pressure region occurs when the solution computed by arc length continuation violates the local constraints (7) and (9) near a cavity end point. This generally corresponds to a fold bifurcation along the solution curve (plotting cavity end point locations against N). N begins to increase again along the curve at such a fold, signalling that the actual solution under further decrease in N is not continuous. I use arc length continuation to extend the solution further until I reach another solution with the value of N at the fold bifurcation, but for which the inequalities (7) and (9) are satisfied. I treat that solution as*

*representing the enlarged cavity that results from decreasing $N$ past the fold bifurcation, and discard solutions computed by arc length continuation that do violate the inequalities (7) and (9).*

*In order to capture the effect of cavity isolation, I compute solutions by arc length continuation under increases in $N$, checking whether inequalities (7) and (9) are satisfied. An isolated cavity forms when the cavity roof contact constraint (7) is violated in the interior of a cavity. In that case, I introduce new contact points where recontact occurs and check whether either of the two new cavities created in the process no longer straddles $P$. Such a cavity is then isolated. I compute its volume, and restart a computation by arc length continuation while imposing equation (8) for that cavity.*

**Referee:** 4. $N^*_{drown}$: On p11, $N^*_{drown}$ is explained clearly, but the meaning of "drown" there (describing separation of ice and bed everywhere across the domain and violation of force balance) seems different from what "drown" describes elsewhere (coverage over a bed protrusion by expanding cavity). Consider using another symbol rather than $N^*_{drown}$?

**Response:** I am afraid I disagree. I think what happens as you pass through $N^*_{drown}$ from above is precisely what "drown" describes elsewhere (coverage over a bed protrusion by expanding cavity): the bed protrusion that used to confine the cavity becomes drowned, as a result of the violation of the compressivity constraint (9).

**Referee:** 5. Some passages consider or refer to bed protrusions or bumps as "less/more prominent" or "smaller/larger (or taller)", e.g. on p17, also elsewhere. Although the literature on this subject may use these adjectives following an agreed convention, or specialists will always understand them, I think you should define their meaning early on. Probably you are comparing maximum elevations, not amplitude (so a "little" short bump superposed onto the peak of a large long bump is the largest/tallest in your analysis), but I am not in fact sure; I may have misinterpreted the meaning.

**Response:** I don't think there is a secret code that the referee hasn't been inducted to here. The intention was purely to be descriptive, since the actual results are contained in the graphical representation of the results in the figures being referenced. I am hesitant, for the sake of the flow of the text, to become too heavy-handed with a formal definition of terminology for what ultimately are illustrative examples. To try to smooth this over with minimal fuss, I have added the following definition-of-sorts to the fifth paragraph of section 3.1:

*(Note that I will use "large" or "prominent" protrusion to describe the protrusion that has the largest difference in height between the local maximum at its top and the local minimum on its upstream side.)*

**Referee:** The meaning of "process-scale/process-scale model" (several times in section 1) is unclear.

**Response:** I do think that 'process-scale" is standard usage in the geosciences to distinguish a model that deals with the first-principles physics of a process from a model applicable at a larger scale, in which the detailed process physics is parameterized. In the context of the description and reference to large-scale models for subglacial drainage in the introduction, the meaning should be clear by context. Yes, I can be explicit about everything I write, but there is a point at which that becomes counterproductive.

**Referee:** p2, line 6, unfinished sentence

**Response.** Thank you. If I read the original text correctly, then there was simply a missing full stop?

**Referee.** p3, line 10, "robustness…obtained to changes". Consider rephrasing this to make it easier to understand.

**Response:** Changed to

*I then investigate in section 3.4 whether changes in bed geometry qualitatively affect the results.*

**Referee:** Fig 1: locate $x = 0$

**Response:** While attractive in principle, this option led to a very cluttered figure when rendered in single column width. Given this is a schematic, $x = 0$ and $x = a$ don't seem particularly essential quantities to visualize, so I've removed the $x = a$ locator instead.

**Referee:** Fig 1 caption: $h(x)$ is cavity roof "elevation" (instead of height)? Line 3, $v$ and $y$ instead of $w$ and $z$ in the partial derivative.

**Response:** Thank you for spotting the ad hoc slip in notation. I've fixed $w$ and $z$ but left "elevation" in the assumption that the reader is able to figure out basic synonyms without my having to spell them out. I hope that's ok.

**Referee:** p4, line 1, missing full stop

**Response:** Thank you.

**Referee:** p5, line 25, "while". Consider using "whereas" or "in contrast" to bring out the contrast against Part 2 better.

**Response:** Replaced with *by contrast*

**Referee:** p6, line 9, "dimensionless combination of effective pressure" doesn't describe eqn. (11)

**Response:** Quite so. I've taken out the "combination of". There had probably been an intitention to say "combiantion of effective pressure and other model parameters", but that seems awkward.

**Referee:** Fig 3a: At first I read Fig 3a as showing the spatial distribution of $N^*$. The horizontal axis needs to be labelled as $b_j^*$, $c_j^*$, as it isn't $x^*$ (although the scale is the same). This issue extends to Fig. 5a to 5b and Fig. 9a to 9d, which are also missing the right axis labels.

**Response:** I'm going to disagree here. The curves show at what value of $x^*$ the various cavity end points are located (as the figure caption states). If I relable the figure axis, you're no doubt going to complain that the bottom panel doesn't show $b^*$ against cavity end point position, at which point I will introduce another set of axis labels below panel (a), different from panel (b), and then it's no longer obvious that the top plot corresponds to the same axis as the bottom plot etc. I think the reader can figure this out from the figure caption rather than the axis label — which you clearly have already. To make this easier, I have labelled the curves for $b_1$, $c_1$, $b_2$ and $c_2$ in panel a with the corresponding symbol (but, to avoid cluttering, I have not repeated that exercise in the other figures identified).

**Referee:** Fig 3 caption, "bed shape $b^*(x^*)$ against $x^{*}$". Change "shape" to "elevation". Also in several other figures.

**Response:** In the same vein as above, having defined what I mean by $b(x)$ previously, in the text and the schematic fig. 1, I'm going to assume that an intelligent reader can figure out that "shape" is used synonymously with "elevation"; it's unclear to me what else "shape" could possibly refer to in this context.

**Referee:** Fig 4 caption, line 1: should $b^*(c^*)$ be $b*(x^*)$? This arises also in the captions of Figures 2 and 7.

**Response:** Correct. It looks like I might have engaged in some careless cutting-and-pasting here.

**Referee:** p8, line 3, "disappear at $N^{*}$". Is something missing here?

**Response:** There indeed is. Changed to $N^* = 0$.

**Referee:** p8, line 6, is the colon before the equal sign a typo?

**Response:** I suspect it wasn't, but used to signify "equals by definition" (which google tells me ":=" has been used to denote since 1894. (I *think* have to define $x_P$.)

**Referee** p9, line 26: "panels (a) and (e) . . . correspond to the same effective pressure". Figure 4 caption gives two different values, 4.01 and 4.02.

**Response:** Well spotted. I think I pulled these solutions from an arc length continuation, so the actual parameter values were not identical. I qualified "the same" with "nearly" in the revised text.

**Referee:** p9, line 33: "bed an only"

**Response:** Changed to "...bed, and only ..."

**Referee:** Fig 6: since you plot two lines, the figure is clearer if you label the y-axis as "N2*, N1*" and add a label next to each line

**Response:** I trust the updated version will do the trick.

**Referee:** Fig 7 caption, line 3: please check whether a minus sign is missing from the second $\sigma_{nn}$

**Reponse:** Yes there was a minus sign missing. Thank you for spotting that.

**Referee:** Fig 8, $h_0$ missing from the $y$-axis label

**Response:** Indeed. Thank you for your (impressively) close reading of detail here.

**Referee:** p14, line 18: trimple — triple

**Response:** Corrected.

**Referee:** Fig 9 caption: lines 8 to 10 have hiccups and duplication in various places. For example, "Panel (c). Panel (e): s abruptly". And "Panel (c)" on line 9 had been introduced on line 6, and I would expect panel (e) (not d) to be covered last.

**Response:** Indeed, there were parts of thie original text I could barely make sense of, and some mislabelling to boot. I have replaced the text with

*Panel (a): effective pressure $N^*$ against cavity end point positions for a fully permeable bed with shape given by equation (15) as solid black curves. Note that the solution is unique. Panel (b): Cavity end point positions for the same bed with a small $P^* = P_b$ centered around $x_P^* = 3.23$ (in the lee of the large bed protrusion, see panel e). Black shows the solution for a single cavity initiated around $x_P^*$. Red shows the solution with a single isolated cavity, blue with two isolated cavities. The dashed black curve show values of $N^*$ at which the single cavity expands abruptly, the dashed red and blue curves show the formation of isolated cavities and the closing of the connected cavity in the presence of one or two isolated cavities. See inset for detail of cavity expansion and formation of an isolated cavity. Panel (c): Cavity end point positions for the same bed with a small $P^* = P_c$ centered around $x_P^* = 5.25$ (in the lee of the smallest bed protrusion as shown in panel e). The dashed line shows the negative value of $N^*$ at which the cavity no longer remains confined and the ice detaches from the bed. Panel (d): Cavity end point positions for the same bed with a small $P^* = P_d$ centered around $x_P^* = 1.03$ (the medium bed protrusion, see panel e). Panel (e): the corresponding bed shape $b^*(x^*)$ defined by equation (15) against $x^*$. The beige strips show the permeable areas $P_b$, $P_c$ and $P_d$ used in panels b–d, respectively.*

**Referee:** Fig 11: (i) in the y-axis label, should $N_2^*$ be $N_1^*$? (ii) Although the gentle slope of the "flat portion" of the blue line is described in the caption and the text, this feature would be much clearer if you set the axes' aspect ratio at 1:1 (natural as both axes are effective pressures). Consider the same for Figure 6.

**Response:** It turns out that the $y$-axis label should have been $N_1^*$, $N_2^*$, $N_3^*$ since $N_2^*$ is also plotted as the black dashed line. I have updated the figure and the caption, the latter as

*Effective pressure in the isolated cavities shown as the blue solution in figure 10b against the corresponding effective pressure $N^* = N_2^*$ in the connected cavity (the equivalent of figure 6 for the triple-bumped bed given by equation (15)). Blue shows the effective pressure $N_1^*$ in the isolated cavity around $x^* = 1.03$, red shows effective pressure $N_3^*$ in the isolated cavity around $x^* = 5.25$, while the black dashed line showsn $N_2^* = N^*$ for the range of values of $N^*$ for which the $j = 2$ cavity is open. Note that $N_1^*$ decreases slightly with forcing effective pressure $N^*$ once the second, smaller isolated cavity around $x^* = 5.25$ has formed.*

I have strethed the vertical axis but not quite to 1:1: there are some economic considerations in play in terms of how much money a figure with a lot of white space will cost me to publish. I work in Canada, where this is a real consideration: I can pay my students a living wage, or I can pay

page charges, but there is no magical fund or separate line item for the latter. For the same reason, I have left figure 6 as is.

**Referee:** p17, line 16: "...depend on the geometry of the bed, and on the parts of the bed...". (i) Clearer if you add "depend" after "and]] or remove the comma. (ii) Clearer if you reword "parts", because the intended meaning is the location/whereabout of the parts, not some undefined properties of parts.

**Response:** I have broken up the sentence and re-written it as

*...at critical values of $N/u_b$. These critical values depend on the geometry of the bed, and on the locations of the parts of the bed that are permeable and therefore intrinsically connected to the ambient drainage system.*

**Referee:** p17, lines 23-26: I don't understand how the implied possibility reported in the final sentence follows logically from the previous passage. Is something explanation missing?

**Response:** I don't think so (in terms of a missing explanation) but I have reworded the penultimate sentence to say

*If drainage system access $P$ is located in the lee of less prominent bed bumps, then (perhaps counterintuitively) connections are made only once sufficiently negative effective pressures are reached, and result in complete ice-bed detachment.*

The point being that you can reach negative effective pressures that are not sufficiently negative to cause ice-bed detachment, and that such negative effective pressures can therefore persist (as advertised in the last sentence of this passage).

**Referee:** p19, line 34: the complication or the definition is being stressed. This sentence is easier to read if written as "The definition of a friction law for an impermeable bed has a second complication that deserves to be stressed"

**Response:** I am not sure I agree: I think the original passage is grammatically unambiguous.

**Referee:** p20, line 23-24: A further limitation that should be mentioned is the limited range of locations $x_P$ chosen in the experiments (on steepest points of lee sides). (Otherwise I think your design of putting $x_P$ on different bumps on the topography works fantastically in illustrating the range of behaviour.)

**Response:** This is true; I'm hesitant to go to town on this here as it seems like a technical point to make in the "Conclusions" (which I assume are supposed to be the condensed insights, simple to read); I am hopeful that the more technical new section 3.3, as the results there don't particularly hint at qualitatively "new" types of behaviour resulting from a shifting of $P$ away from the uncavitated normal stress minima.

**Referee:** p21, first line of A1, spurious comma after 54

**Response:** corrected

**Referee:** p21, line 23, "whree"

**Response:** ditto

**Referee:** Eq (A3): the integrand on the left should be a sum instead of difference. Capital $C$ in $h'_c$, and $b''$ should be $b'$.

**Response:** Thank you for fixing those. My rather sad suspicion is that you are the first and last person to actually bother reading this bit, but for that alone, it was worth writing.

**Referee:** p23, line 15, hiccup after $G(\infty)$

**Response:** ")" replaced by "0"

**Referee:** p24, line 2, "!98"

**Response:** I assume I must have been excited about page 98 of my PhD thesis when I wrote this originally, but I will reluctantly agree to removing the exclamation mark.

**Reviewer** p24, line 5, to help readers, it is useful to (re)qualify $V_j$ as cavity water volumes here. Again, capital $C$ in $h'_c$ ?

**Response:** Yes to both.

**Reviewer:** p24 last line, full stop before "To"

**Response:** I must have fixed that somewhere in the interim; my local copy of the file already has a full stop in place.

---

## Author Comment (AC2)

**The evolution of isolated cavities and hydraulic connection at the glacier bed. Part 1: steady states and friction laws**

Christian Schoof

May 22, 2023

**1 Referee #2**

**Referee:** - In this work, once a cavity becomes hydraulically isolated, the model enforces that the cavity have constant volume until it becomes hydraulically connected once again. Under these conditions, the water pressure of the cavity is an unknown to compute. Something that came to mind is the possibility of hydraulically isolated cavities that develop an air-filled gap in between the water and the ice. I can understand the reasoning behind the constant volume condition when, e.g., the far-field ice speed decreases, since the cavity would want to close but it cannot compress the water in it. However, upon an increase in this far-field speed, I find it logical that, at a certain point, when the water pressure becomes sufficiently low (perhaps 0?), a partially air-filled cavity starts to develop. In other words, it is reasonable to assume that a hydraulically isolated cavity is still pneumatically connected, since the opposite sounds unrealistic for most glacier conditions. In the context of your model, allowing the formation of air-filled cavities might mean enforcing pneumatic connectedness when a low water pressure is reached, and therefore allowing the volume to increase beyond the water volume. I mention this point because it would mean that the unbounded increase in basal stress that you see for the single cavity in Figure 8 would perhaps no longer hold. If this comment is sensible, I think it should be commented in the discussion.

**Response:** Thank you for pointing this out; a good point, which I would rather not have dealt with (being lazy, as it makes the model more complicated!) — in particular, it increases the size of the parameter space by one, since $N$ and overburden $p_i$ are independent parameters, and it is really $p_i$ that controls when the type of cavities described should form. I have amended the text after equation (9) to describe how the model would need to be changed if one were to deal with the complication raised: *Outside of the intervals $(b_j - \delta, b_j)$ and $(c_j, c_j + \delta)$, (9) can, and in general will, be violated somewhere as indicated in figure 1. The possibility of such underpressurized regions is the primary difference between the permeable and impermeable bed models. By not bounding compressive normal stress everywhere at the bed, the model does however not allow for vapour-filled cavity to form if the normal stress dropping to the triple-point pressure of water. In order to incorporate the latter effect, I would need to add the constraint that $p - 2\eta \partial v / \partial y > -p_i$ in $C'$, where $p_i$ s ice overburden, and set $p - 2\eta \partial v / \partial y = -p_i$ in any cavity that does not straddle $P$ and in which the prescribed water volume $V_j$ (potentially equal to zero) would lead to an effective pressure $N_j > p_i$ if the volume constraint \*8) were imposed. I omit this complication here on the basis that I expect overburden $p_i$ to be large compared with the typical normal stress variations caused by ice flow over bed undulations; suffice it to point out that the model described in part 2 can in principle describe vapour-filled cavities,*

Note that I refer to vapour-filled cavities: I don't believe that the glacier is actally likely to be pneumatically-connected in the sense that air can flow but water cannot (from personal experience, I can attest to the existence of pressurized air-filled pockets that can expel a hot water drill head when punctured; such air pockets presumably should not exist if pneumatic connection in the strict sense is maintained). That does not detract from the possibility of cavities that are not filled with liquid water as described in the exceprt from the updated paper.

As stated above, I have not tried to implement the possibility of "dry crevasses" in the steady state model. The ability to describe fluid lag does mean the model in part 2 can, in principle, describe such dry crevasses, though it would require setting the overburden to much lower values than I have tried in that paper.

With regard to preventing unbounded basal drag, I have added the following paragraph at the end of section 3.2:

*As a further caveat, note that for a fixed $N$, unbounded $\tau_b$ as shown in figure 8 results from the ability to generate arbitrarily large compressive normal stresses on the upstream side of the smaller bed protrusion, balanced by correspondingly negative compressive normal stresses on the downstream side in the hydraulically isolated low pressure region on the downstream side of the larger protrusion (figure 7c2, where $-\sigma_{nn}^*$ is scaled with $1/u_b$, so the actual stress is the pattern shown multiplied by a coefficient proportional to $u_b$). As described in section 2 immediately after equation (9), arbitrarily negative normal stresses cannot be generated since a vapour-filled cavity will eventually form, and this should lead ultimately to a bounded basal drag satisfying an amended version of Iken's bound, $\tau_b \leq \max(\partial b/\partial x)p_i$, where $p_i$ is once more overburden; the model here ignores that possibility, effectively treating $p_i$ as infinite for the purposes of bounded basal drag.*

**Referee:** p3, line 18: Consider starting sentence with a non-mathematical symbol.

**Response:** Added "Here, ..."

**Referee:** p3, line 23: "the normal component of velocity vanishes" ¿ "the normal component of the velocity vanishes".

**Response:** Is that change strictly necessary, grammatically speaking? I'm inclined to say it is not.

**Referee:** p 5, line 14: In Stubblefield et al. (2021), the rate of change of the volume is enforced, not the total cavity volume. Mathematically, this is carried out by setting the integral of the normal velocity at the lower boundary equal to this prescribed rate of change. The Lagrange multiplier associated to this constraint is the water pressure.

**Response:** If I constrain the rate of change of volume to be zero (which I believe is what Stubblefield et al do), then am I not constraining the volume to remain constant? I understand that there is a difference in terms of how this is implemented, but the result is the same. In any case, I've changed the wording to

*...Instead, the total cavity volume is prescribed through initial conditions, the constraint itself being imposed on normal velocity so as to conserve that initial volume.*

**Referee:** p5, line 16: As you write, I also find the specification of a permeable point $x_P$ along the bed awkward, from a physical point of view. It seems to me that the fact that you choose it to be the location where cavities first form is effectively equivalent to choosing that a particular cavity, once it forms, is connected to the external drainage system. I think it would be helpful to the reader to clarify the intention behind choosing $x_P$ to be this point.

**Response:** I think this is the same point as raised by the other referee, in slightly different form. I have reworded the section flagged above as

*Below, I will typically consider either the entire bed permeable with $P = (0, a)$, or I will consider a small permeable patch around a single location, which I will denote by $x_P$. I will typically choose $x_P$ to be the location of a local minimum of compressive normal stress for an uncavitated bed, since*

*that is where cavities first form for a permeable bed. In addition, in section 3.3 I consider larger permeable bed portions P that do not align with these normal stress minima.*

Depending on your take, there may not be a very good reason for locating $x_P$ as described, especially if we read the model description literally, so $P$ is a (geologically-determined) permeable part of the bed. If we suppose that $P$ is a proxy for lateral drainage access across an unmodelled of ice-bed interface that lies to the side of the flowline that *is* modelled, then it may make more sense to consider $x_P$ at normal stress minima.

That being said, I have introduced a new section 3.3 to investigate what happens if you locate $P$ elsewhere. The start of that section reads

*The results above were computed either for completely permeable beds, or for beds that had permeable sections located at normal stress minima prior to cavity formation. As pointed out, I view these permeable bed portions P potentiallly as proxies for lateral access from a three-dimensional ambient draiange system along an unmodelled part of the ice-bed interface, to one side of the flowline that model describes. In that case it may make sense for that laeral access to reach the modelled flowline in places where compressive normal stress has local minima. Locating the permeable where cavities form at the highest possible values of N is also convenient as it reduces the number of additional parameters that describe the bed in the absence of a more sophisticated three-dimensional model.*

*In order to investigate the effect of choosing different permeable bed portions P, I plot in figure 9 the dependence of cavity end point positions on effective pressure $N^*$ for the same bed geometry (equation (10) as before, but for two alternative choices of $P^*$, ...*

I won't reproduce the entire text of that section here, please see the revised paper for details.

**Reviewer:** p 6, line 29: "by plotting cavity end point position" $\mapsto$ "by plotting the cavity end point positions"

**Response:** corrected

**Reviewer:** p 8, figure 4: I think it would be helpful to the reader to include a subtle line in the plots for the normal stress that indicates the values of $N^*$. It would make it much easier to notice the areas of low pressure and the difference in water pressure between isolated and connected cavities. Same for figure 7.

**Response:** Done

**Reviewer:** p 9, line 3: "see also fig 3" $\mapsto$ "see also figure 3"

**Response:** Done

**Reviewer:** p 9, line 9: "decreased." $\mapsto$ "decreased"

**Response:** full stop replaced by comma

**Reviewer:** p9, line 33: "we start with an uncavitated bed an only lower $N^*$", this sentence does not make sense.

**Response:** Replaced with "We start with an uncavitated bed, and only lower $N^*$"

**Reviewer:** p10, Figure 5: Close brackets in line 3 of caption.

**Response:** Done.

**Reviewer:** p11, line 6: "cavity cavity" $\mapsto$ "cavity"

**Response:** oops. . . corrected.

**Reviewer:** p13, line 5: "by arrow" $\mapsto$ "by the arrow"

**Response:** Done

**Reviewer:** p13, line 12: Consider starting sentence with a non-mathematical symbol.

**Response:** Added "Drag"

**Reviewer:** 'p14, line 19: "trimple" $\mapsto$ "triple"

**Response:** Done

**Reviewer:** p16, line 7: "is" $\mapsto$ "are"

**Response:** Done.

**Reviewer:** p16, line 8: "again be unbounded" $\mapsto$ "again unbounded"

**Response:** Done.

**Reviewer:** p17, line 30: close brackets.

**Response:** Done.

**Reviewer:** p17, line 32: "it easier" $\mapsto$ "it is easier"

**Response:** Corrected.

**Reviewer:** p18, line 20: "constraint" $\mapsto$ "constraints"

**Response:** Done.

**Reviewer:** p18, line 33: "If drainage system access" $\mapsto$ "If the drainage system access"

**Response:** Again, I'm going to disagree on the need to use of a definite article here, on the basis that drainage system access is not countable.

**Reviewer:** p18, line 34: "effective pressure" $\mapsto$ "the effective pressure"

**Response:** Ditto.

**Reviewer:** p19, line 18: "bed," $\mapsto$ "bed."

**Response:** Corrected.

---

## Author Response (AR2)

**The evolution of isolated cavities and hydraulic connection at the glacier bed. Part 1: steady states and friction laws**

Christian Schoof

August 27, 2023

There are no referee comments to respond to.